

# Exact diagonalization, matrix product states and conformal perturbation theory study of a 3D Ising fuzzy sphere model

Andreas M. Läuchli[1⋆∘], Loïc Herviou[2†], Patrick H. Wilhelm[3] and Slava Rychkov[4‡]

**1** Institute of Physics, École Polytechnique Fédérale de Lausanne (EPFL),
1015 Lausanne, Switzerland
**2** Univ. Grenoble Alpes, CNRS, LPMMC, 38000 Grenoble, France
**3** Institut für Theoretische Physik, Universität Innsbruck, A-6020 Innsbruck, Austria
**4** Institut des Hautes Études Scientifiques, 91440 Bures-sur-Yvette, France

⋆ andreas.laeuchli@epfl.ch , † loic.herviou@lpmmc.cnrs.fr , ‡ slava@ihes.fr

## Abstract

Numerical studies of phase transitions in statistical and quantum lattice models provide crucial insights into the corresponding Conformal Field Theories (CFTs). In higher dimensions, comparing finite-volume numerical results to infinite-volume CFT data is facilitated by choosing the sphere $S^{d-1}$ as the spatial manifold. Recently, the fuzzy sphere regulator [1] has enabled such studies with exact rotational invariance, yielding impressive agreement with known 3D Ising CFT predictions, as well as new results. However, systematic improvements and a deeper understanding of finite-size corrections remain essential. In this work, we revisit the fuzzy sphere regulator, focusing on the original Ising model, with two main goals. First, we assess the robustness of this approach using Conformal Perturbation Theory (CPT), to which we provide a detailed guidebook. We demonstrate how CPT provides a unified framework for determining the critical point, the speed of light, and residual deviations from CFT predictions. Applying this framework, we study finite-size corrections and clarify the role of tuning the model in minimizing these effects. Second, we develop a novel method for extracting Operator Product Expansion (OPE) coefficients from fuzzy sphere data. This method leverages the sensitivity of energy levels to detuning from criticality, providing new insights into level mixing and avoided crossings in finite systems. Our work also includes validation of CPT in a 1+1D Ising model away from the integrable limit.

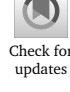

## Contents

---

∘Note that significant part of this work was conducted while the first author was in a role at PSI, Villigen, Switzerland.

# 1  Introduction

Numerical studies of continuous phase transitions in statistical and quantum lattice models are a valuable source of information about the corresponding Conformal Field Theories. In numerical studies done in finite volume, extracting infinite-volume CFT data may be more or



less difficult depending on the choice of the spatial manifold. For 1+1D models, studies of quantum Hamiltonians and transfer matrices on a circle $S^1$ allow for easy comparison to CFT thanks to the radial quantization [2–5]. For higher-dimensional models, analogous simplicity would arise for the spatial manifold being the sphere $S^{d-1}$ [6]. This was studied for 3D lattice models [7–12], but recovering rotational invariance of the sphere remained a challenge.

Recently, Ref. [1] achieved significant progress in this direction via the fuzzy sphere regulator. In a nutshell, they considered a Hamiltonian describing a finite number $N$ of electrons on the surface of $S^2$ moving in a constant normal magnetic field and interacting with a short-range potential. Crucially, the model is exactly rotational invariant. Varying the potential, they tuned the system to a quantum phase transition which, in the $N \to \infty$ limit, is in the 3D Ising universality class. The spectrum of the Hamiltonian at criticality was then compared to the 3D Ising CFT, well known due to the conformal bootstrap [13], showing impressive agreement even for modest values of $N$, and also extracting some energy levels not yet accessed by the bootstrap. Since then, the fuzzy sphere regulator was used to study other Ising CFT observables [14–23] and phase transitions in other universality classes [24–28].

While the results of Ref. [1] represent an excellent baseline for the new fuzzy sphere method, more work is needed to assess how successful this model actually is and to find the best strategy for systematic improvements. This is the first purpose of our work. Here we will be focusing on the original 3D Ising fuzzy sphere model of [1], but our considerations are general and can be applied to other models.

We will be relying on Conformal Perturbation Theory (CPT), recently advocated to describe 2+1D models close to but not exactly at their quantum critical point [29,30].[1] We will see that CPT provides a comprehensive framework to address many issues arising when comparing a fuzzy sphere model to a CFT, such as: to robustly locate the critical point of the model; to determine the speed of light; to parametrize residual deviation between the microscopic model spectrum and the CFT through an effective theory; or to understand why some models belonging to the same critical line agree with CFT better than others. In particular, we will elucidate what is special about the choice of $V_0 = 4.75$ in [1].

With the help of CPT, we will be able to understand and partially subtract finite size corrections, improving the agreement of the fuzzy sphere with the CFT data precisely known thanks to the conformal bootstrap. However, from looking at a larger array of data than what was unveiled in [1], we will see that finite size corrections do remain significant for some energy levels.

The second purpose of our work will be to develop a novel method for extracting OPE coefficients of the underlying CFT from the fuzzy sphere model data. Previously, OPE coefficients were extracted [14] from a matrix element of a microscopic operator interpolating a CFT operator between eigenstates of the Hamiltonian. Here instead we will note that OPE coefficients of the most relevant $\mathbb{Z}_2$ even CFT operator $\varepsilon$ can be extracted by studying the variation of the eigenenergies when the model is detuned from the critical point. We will see that this gives a rather robust scheme. In addition, this allows us to highlight interesting level mixing and repulsion effects which complicate the interpretation of fuzzy sphere data at currently attainable volumes.

We start in Sec. 2 by reminding the reader what is known about the Ising CFT in 2D where it is exactly solved and in 3D thanks to the numerical conformal bootstrap. We also review here what happens when a CFT is transferred from the flat Euclidean space $\mathbb{R}^d$ to the geometry of the cylinder $S^{d-1} \times \mathbb{R}$, as appropriate for comparing with simulations of quantum models on $S^{d-1}$. In Sec. 3 we review the method of CPT for describing perturbations of energy levels on $S^{d-1}$ when the CFT is perturbed by small relevant or irrelevant interactions.

---

[1]This framework also proved useful in the recent fuzzy sphere studies [22,31].

Then, in Sec. 4 we test the CPT method in 1+1D. Here we study a 1+1D quantum Hamiltonian with a critical point in the 2D Ising CFT universality class. For a fair comparison, we choose the Hamiltonian not to be exactly solvable. Working on spin chains of length up to $N = 26$, we determine the critical point and the speed of light via the CPT method, and we proceed to study residual deviations from the CFT using an effective field theory with three couplings. We find excellent agreement with the CPT predictions derived long ago by Reinicke [32, 33].

In Sec. 5 we present the numerical results for the fuzzy sphere model in 2+1D and analyze them using the CPT method. Our analysis here differs from Ref. [1] in several important details, such as:

- The critical point is determined not through the scaling of the order parameter but by looking at where the CPT coupling $g_\varepsilon$ crosses zero [29].

- While Ref. [1] privileged the stress tensor level, rescaling the whole spectrum so that the stress tensor scaling dimension is exactly 3, we note that the energy levels $\sigma$ and $\partial\sigma$ are the least affected by perturbations, and fix those levels to their conformal bootstrap values to determine $g_\varepsilon$ and the speed of light.

We first study the fuzzy sphere model at the fine-tuned value $V_0 = 4.75$ of the Haldane pseudopotential coefficient found in Ref. [1] to minimize corrections to scaling. Showing the dependence of energy levels on the system size, we illustrate that in some sectors finite size corrections are significant for attainable system sizes, even for this $V_0$. We also study the model at $V_0 = 2.5$ and $V_0 = 6$, to understand better what is special about $V_0 = 4.75$. We found that the two most important irrelevant CPT couplings $g_{\varepsilon'}$ and $g_C$ are smaller at $V_0 = 4.75$ than away from this value. (The same conclusion for $g_{\varepsilon'}$ was reached in [22].) In addition, there is no finite-size drift of the critical point at $V_0 = 4.75$, which we hypothesize is related to the vanishing of a curvature contribution to the $g_\varepsilon$ coupling (Sec. 5.5.1).

In Sec. 6 we develop our method for extracting a subset of OPE coefficients, having the form $f_{\mathcal{O}\mathcal{O}\varepsilon}$. The idea here is that the variation of the energy level corresponding to the CFT operator $\mathcal{O}$ when one slightly detunes the model away from criticality, is proportional to this OPE coefficient. So the OPE coefficient can be determined by taking the derivative of the energy level (corrected for the speed of light) with respect to the CPT coupling $g_\varepsilon$. We show that this procedure works very well. Of course there are still finite size corrections, but the needed extrapolations to infinite volume are often small. We use this technique to show agreement with the known OPE coefficients, and report some new ones. This analysis also illuminates some avoided energy level crossings which are visible in the energy spectrum as one varies the size of the system. In fact, in an avoided level crossing, the identity of states changes, and this reflects in a large excursion of the OPE coefficients.

In Sec. 7, which lies a bit away from the main line of development, we try to understand qualitatively (and only with partial success) the level mixing effects and the avoided level crossings observed the spectrum.

In Sec. 8 we conclude.

Several detailed discussions are relegated to the appendices. In App. A we provide a guide to the 3D Ising CFT data used in our work. In App. B we describe calculations of relative factors for CPT corrections of descendant states with respect to primary states, and other CPT details. The derivations of all the reported results can be found in the accompanying Mathematica notebook [34]. App. C reviews the 1+1D CPT results of Reinicke [32, 33]. App. D describes the fuzzy sphere model, and App. E the numerical methods used in our work: Exact Diagonalization (ED) for smaller system sizes and Matrix Product States (MPS) for larger ones. We have not used the excellent numerical package FuzzifiED.jl [35] but relied on our own codes.

**Notation.** In this paper $d$ denotes the full spacetime dimension, so $d = 3$ for the 2+1D Ising model.

# 2 Ising CFT basics

## 2.1 2D

We start by recalling the main features of the 2D Ising CFT, which is an exactly solved theory having Virasoro conformal symmetry of central charge $c = 1/2$ [36]. The theory is unitary, and is invariant under global $\mathbb{Z}_2$ spin parity and under spatial parity. There are three Virasoro primary local operators $\mathbb{1}$, $\sigma$, $\varepsilon$, of conformal weights $h = \bar{h} = 0, 1/16, 1/2$. We remind the reader that the scaling dimension of the operator is $\Delta = h + \bar{h}$ and its conformal spin is $s = h - \bar{h}$. Acting on them by a string of Virasoro raising generators $L_k$ and $\bar{L}_{k'}$ with $k, k' < 0$, fills three conformal multiplets, containing all other local operators $\Phi_{h,\bar{h}}$ of the theory, whose conformal weights differ from primaries by an integer. Important low-lying local operators include the stress tensors $T_{2,0}$ and $\bar{T}_{0,2}$, the leading irrelevant scalar $(T\bar{T})_{2,2}$ and the spin $\pm 4$ operators $(T^2)_{4,0}$ and $(\bar{T}^2)_{0,4}$.[2]

An important feature of any 2D CFT is that it can be equivalently considered on the infinite flat space $\mathbb{R}^2$ or on the "cylinder" $S_R^1 \times \mathbb{R}$, where $S_R^1$ is a circle of radius $R$. The map from one geometry to the other is obtained by applying a logarithmic conformal map. In the latter description, local operators inserted at the origin of $\mathbb{R}^2$ become states in the Hilbert space of the theory on the circle $S_R^1$, while the CFT dilatation operator becomes the Hamiltonian, evolving states in the (Euclidean) time direction along the cylinder. The energies and momenta of states on the circle are related to scaling dimensions and spins of local CFT operators:

$$E_i^{\text{CFT}} = \frac{1}{R}(\Delta_i - c/12), \qquad P_i^{\text{CFT}} = \frac{s_i}{R}. \tag{1}$$

What happens to these continuum relations in a discretized description? Suppose we have a critical quantum spin chain Hamiltonian acting on $N$ quantum spins with periodic boundary conditions. We identify $N$ with the length of the circle, i.e. $R = N/2\pi$. Suppose that the phase transition is described by a CFT, in particular the dynamical critical exponent $z = 1$. Then, for $N \gg 1$, the energies of the low-lying Hamiltonian eigenstates will be related to the CFT energies from (1) by

$$E_i \approx v E_i^{\text{CFT}} + \rho_0 N, \tag{2}$$

where $\rho_0$ is the microscopic ground-state energy density and $v$ is the speed of light. While not predicted by CFT, they can be determined from a fit. The central charge, scaling dimensions and spins of local CFT operators can then be extracted numerically. This is the essence of the finite-size scaling method for 2D CFTs [5]. Note that since momentum is quantized, it receives no corrections: $P_i = P_i^{\text{CFT}}$.

## 2.2 3D

We next review the 3D Ising CFT [37, Sec. B.2], which will be the focus of our study using the fuzzy sphere regulator. As in 2D, it is a unitary theory with a global $\mathbb{Z}_2$ spin parity and a spatial parity. In 3D there is no Virasoro algebra, as the conformal group in $d \geq 3$ is finite dimensional [36]. We have global conformal primaries $\mathcal{O}_{\Delta,\ell}$, where $\Delta \geq 0$ is the scaling dimension and $\ell \in \mathbb{Z}_{\geq 0}$ is the spin. They are $\ell$-index symmetric traceless tensors, transforming in a $2\ell + 1$-dimensional representation under SO(3) rotations. The rest of operators organize in global conformal multiplets obtained by acting on primaries with derivatives; these are the descendants.

---

[2] $T\bar{T}$, $T^2$ and $\bar{T}^2$ are the common notation for the operators $L_{-2}\bar{L}_{-2}\mathbb{1}$, $L_{-2}^2\mathbb{1}$ and $\bar{L}_{-2}^2\mathbb{1}$.

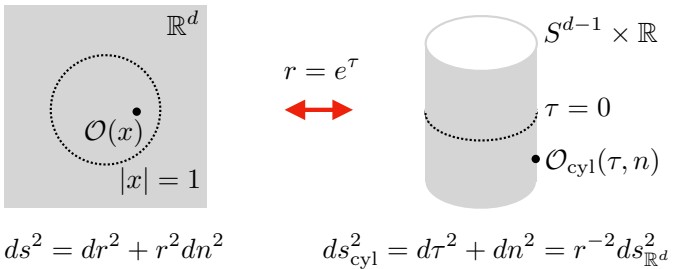

$$ds^2 = dr^2 + r^2 dn^2 \qquad ds_{\text{cyl}}^2 = d\tau^2 + dn^2 = r^{-2} ds_{\mathbb{R}^d}^2$$

Figure 1: Weyl transformation from $\mathbb{R}^d$ to the cylinder $S^{d-1} \times \mathbb{R}$.

The 3D Ising CFT is not exactly solved. Only the unit operator $\mathbb{1}_{0,0}$ and the stress tensor $T_{3,2}$ have exactly known dimensions; the rest have to be studied numerically. The number of primaries is infinite; the spectrum is dense at high scaling dimensions but sparse at low scaling dimensions. There is also an interesting structure in the spectrum of operators of high spin and low twist $\tau = \Delta - \ell$, which form regular families of "double-twist operators" [13], whose existence may be understood via analytic continuation in spin [38].

The 3D Ising CFT spectrum at low scaling dimension, as well as at high spin but low twist, is known with good accuracy thanks to the advances in the numerical conformal bootstrap [13, 39–45]. This includes most importantly the relevant $\mathbb{Z}_2$-odd scalar $\sigma$ and the relevant $\mathbb{Z}_2$-even scalar $\varepsilon$ of dimensions [45]

$$\Delta_\sigma = 0.518148806(24), \qquad \Delta_\varepsilon = 1.41262528(29). \tag{3}$$

It may be sometimes helpful to think of $\sigma$ and $\varepsilon$ as of renormalized versions of the operators $\phi$ and $\phi^2$, where $\phi$ is the field in the Landau-Ginzburg description of the Ising model critical point. Further accurately-known operators are the leading irrelevant scalar $\varepsilon'$, the leading spin-4 primary $C$, etc. OPE coefficients of many of these operators are also known. The 3D Ising CFT data used in our work are summarized in App. A.

## 2.3 $\mathbb{R}^d$/cylinder correspondence

Conformal bootstrap usually studies $d$-dimensional CFTs on $\mathbb{R}^d$. Another preferred geometry is the cylinder $S^{d-1} \times \mathbb{R}$. In fact, any CFT quantity can be transferred back and forth between $\mathbb{R}^d$ and the cylinder, because these two geometries are related by a Weyl transformation of the metric.[3] We will next review this correspondence, which will be very important for our work, following [46, 47].

Fig. 1 illustrates the Weyl transformation between $\mathbb{R}^d$ and the cylinder. Primary[4] CFT operators on the two manifolds are related by rescaling factors:

$$\mathcal{O}_{\text{cyl}}(\tau, n) = r^{-\Delta_\mathcal{O}} \mathcal{O}(x), \qquad x = rn, \quad r = e^\tau, \quad |n| = 1. \tag{4}$$

The meaning of this equation is that the CFT correlators on the cylinder are obtained by rescaling the correlators on $\mathbb{R}^d$:

$$\langle \mathcal{O}_{\text{cyl}}(\tau, n) \ldots \rangle = e^{-\tau \Delta_\mathcal{O}} \langle \mathcal{O}(x = e^\tau n) \ldots \rangle. \tag{5}$$

The so-defined correlators on the cylinder are $\tau$-translation invariant, as they should be. We stress that Eq. (5) is valid for primaries. Relations between correlators of descendants are obtained by differentiating this equation.

---

[3] This general $d$ argument takes place of the log map in $d = 2$.

[4] Here and in the rest of the paper "primary" means primary under the global conformal group. The other Virasoro primaries in 2D will be always referred to in full as "Virasoro primaries." The term "quasiprimary" will not be used. This is done to uniformize terminology between 2D and higher $d$.

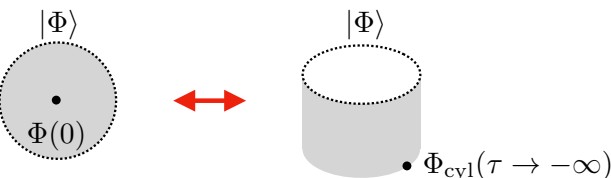

Figure 2: Preparing ket states on $\mathbb{R}^d$ (left) and on the cylinder (right).

Applying the usual cutting and gluing logic to the path integral [48], we can also think of CFT correlators quantum mechanically, both on $\mathbb{R}^d$ and on the cylinder (Fig. 2). This point of view will be central for our paper. On $\mathbb{R}^d$, one works in the so called "radial quantization," where the CFT states live on spheres centered at the origin, with the CFT dilatation operator $D$ playing the role of the Hamiltonian. In this picture, inserting the CFT operator $\Phi(0)$ (primary or descendant)[5] at the origin prepares the ket state $|\Phi\rangle$ living on the unit sphere, which is an eigenstate of $D$:

$$D|\Phi\rangle = \Delta_\Phi |\Phi\rangle. \tag{6}$$

These states on the sphere transform under SO($d$) rotations as the corresponding CFT operators. In 3D, they therefore form spin-$\ell_i$ irreps. On the cylinder, the same CFT states live on the spheres $S^{d-1}$ at constant $\tau$, with $\tau = 0$ being the image of the unit sphere in $\mathbb{R}^d$. The same ket state at $\tau = 0$ can be produced by inserting appropriate cylinder operators $\Phi_{\text{cyl}}(\tau = -\infty)$ in the infinite past, although we will not need their exact expressions.[6]

When we map $\mathbb{R}^d$ to the cylinder via $r = e^\tau$, the dilatation vector field $r\partial_r$ become the translation along the cylinder $\partial_\tau$, the dilatation generator $D$ is mapped to the $\tau$-translation generator which is precisely the CFT Hamiltonian on the sphere $H_{\text{CFT}}$, and the scaling dimensions becomes the energies of the corresponding states, up to a constant shift $w$ which is the Casimir energy due to the Weyl anomaly:

$$H_{\text{CFT}}|\Phi\rangle = (\Delta_\Phi + w)|\Phi\rangle. \tag{7}$$

We saw in (1) that $w = -c/12$ in 2D; see [49] for the 4D case. In 3D as in all odd $d$ there is no Weyl anomaly and we have $w = 0$.

We will also need the corresponding bra states, which are obtained by inserting "reflected" operators $\Phi^\dagger$ at infinity in $\mathbb{R}^d$ and at infinite future on the cylinder (Fig. 3). The reflection acts on coordinates as $\tau \to -\tau$ on the cylinder, and as inversion $\mathcal{I} : x_\mu \to x_\mu/x^2$ on $\mathbb{R}^d$. Reflection action on operators is first specified on primaries, and then extended to descendants by linearity. On scalar primaries, reflection acts as[7]

$$[\mathcal{O}_{\text{cyl}}(\tau, n)]^\dagger = \mathcal{O}_{\text{cyl}}(-\tau, n), \qquad (\text{cylinder}), \tag{8}$$

$$[\mathcal{O}(x)]^\dagger = |x|^{-2\Delta_\mathcal{O}} \mathcal{O}(\mathcal{I}x), \qquad (\mathbb{R}^d). \tag{9}$$

On tensor primaries, the only difference is that one "flips" indices in the $\tau$ direction on the cylinder and in the radial direction on $\mathbb{R}^d$. We give an example for spin-1 primaries, the extension for more indices being straightforward:

$$[(\mathcal{O}_\mu)_{\text{cyl}}(\tau, n)]^\dagger = \Theta^{\text{cyl}}_{\mu\nu}(\mathcal{O}_\nu)_{\text{cyl}}(-\tau, n), \qquad (\text{cylinder}), \tag{10}$$

$$[\mathcal{O}_\mu(x)]^\dagger = |x|^{-2\Delta_\mathcal{O}} \Theta_{\mu\nu} \mathcal{O}_\nu(\mathcal{I}x), \qquad (\mathbb{R}^d), \tag{11}$$

---

[5]We will denote by $\Phi$ a generic CFT operator which may be a primary or a descendant. We will reserve curly letters like $\mathcal{O}$, $\mathcal{V}$ etc for primaries.

[6]For $\Phi = \mathcal{O}$ primary we have $\mathcal{O}_{\text{cyl}}(\tau = -\infty) = \lim_{\tau \to -\infty} e^{-\tau\Delta_\mathcal{O}} \mathcal{O}_{\text{cyl}}(\tau, n)$. For $\Phi$ descendant, $\Phi_{\text{cyl}}(\tau = -\infty)$ has to be obtained by differentiating (5).

[7]Note that in this paper we work with real operators. For complex operators one one would have to add complex conjugation of the operator to the reflection map.

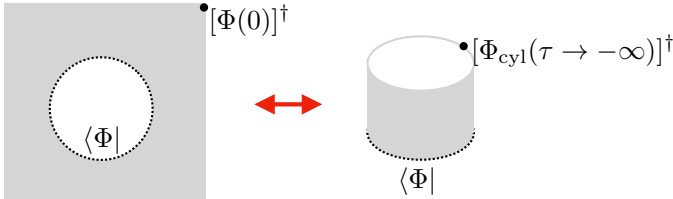

Figure 3: Preparing bra states on $\mathbb{R}^d$ (left) and on the cylinder (right).

where $\Theta^{\text{cyl}}$ is a diagonal matrix with $-1$ as the $\tau\tau$ entry, and 1 for the indices along the sphere, while on $\mathbb{R}^d$ we have $\Theta_{\mu\nu} = \delta_{\mu\nu} - x_\mu x_\nu / x^2$.

Eq. (7) determines the spectrum of the CFT Hamiltonian on the sphere $S^{d-1}$ of unit radius. When we work on the sphere of radius $R$, the spectrum is rescaled accordingly:

$$E_i^{\text{CFT}} = \frac{\Delta_i + w}{R}. \tag{12}$$

Now suppose we have a finite microscopic Hamiltonian on the sphere which realizes in the limit of an infinite number $N$ of degrees of freedom a quantum phase transition whose universality class is that of a CFT. $N$ could be a number of spins distributed on the sphere [29]. For the fuzzy sphere, $N$ is the number of available magnetic orbitals, not taking into account the spin degeneracy. For the model we consider (see App.D for more details), it also corresponds to the number of electrons. The number of orbitals is directly proportional to the area of the sphere $S_R^{d-1}$, such that its radius $R \propto N^{1/(d-1)}$. This corresponds to a physical limit of constant magnetic field on the surface of the sphere when scaling $N$. In such a setup, we expect that the microscopic energy levels on the sphere will be related to the CFT energy levels (12) by the same Eq. (2) as in the 2D case.

## 3 Conformal perturbation theory

We will next review the effective field theory (EFT) which will allow us to quantify the deviation between microscopic and exact CFT energy levels, i.e. the error in (2). We assume that the microscopic theory has a local Hamiltonian, as is true for 1+1D spin chains and for 2+1D fuzzy sphere regulator models. The basic RG intuition suggests that any local theory close to a CFT can be described by an EFT perturbing the CFT by a collection of local operators [50, 51].

### 3.1 Corrections to correlation functions in $\mathbb{R}^d$

Let us first briefly discuss what happens in $\mathbb{R}^d$. After rescaling the Euclidean time to set the speed of light to one, the appropriate EFT action would take the form

$$S_{\text{EFT}} = S_{\text{CFT}} + \Delta S, \qquad \Delta S = \sum_{\mathcal{V}} G_{\mathcal{V}} \int_{\mathbb{R}^d} d^d x \, \mathcal{V}(x), \tag{13}$$

where $\mathcal{V}$ may be any CFT operator allowed by the microscopic symmetry (excluding total derivatives as having no effect on a manifold without a boundary), and $G_{\mathcal{V}}$ are the couplings. They depend on the microscopic theory and in practice their values are unknown. We assume that the perturbation arises at the microscopic distance scale $a = 1$, a short distance cutoff which sets our unit of length. Conformal perturbation theory [50–55] is a well-known method for evaluating correlators starting from (13). The idea is to insert $e^{-\Delta S}$ inside the CFT

correlators, expand the exponential, and evaluate the resulting integrals in $x$. For example, the two point function of a field $\Phi$ would be evaluated in this approach as:

$$\langle \Phi(0)\Phi(r)\rangle_{\text{EFT}} = \langle \Phi(0)\Phi(r)e^{-\Delta S}\rangle_{\text{CFT}}$$

$$= \langle \Phi(0)\Phi(r)\rangle_{\text{CFT}} - \sum_{\mathcal{V}} G_{\mathcal{V}} \int_{\mathbb{R}^d} d^d x \langle \Phi(0)\Phi(r)\mathcal{V}(x)\rangle_{\text{CFT}} + \dots \qquad (14)$$

The resulting corrections to CFT correlators behave differently depending whether the operator $\mathcal{V}$ is relevant or irrelevant. Perturbations from irrelevant $\mathcal{V}$ decay with the distance, while the corrections due to relevant $\mathcal{V}$ grow and remain small at distances $x \ll \xi$, where $\xi$ is the correlation length. Being near CFT in this context means that $\xi \gg 1$. This condition is equivalent to $G_{\mathcal{V}} \ll 1$ for the relevant couplings.[8] The correlation length can be estimated from

$$G_{\mathcal{V}}\xi^{d-\Delta_{\mathcal{V}}} \sim 1 \qquad (\mathcal{V} \text{ relevant}). \qquad (15)$$

In practice, the smallness of the relevant couplings is achieved by tuning the microscopic model to the vicinity of the critical point. The irrelevant operators may and will in general have couplings $G_{\mathcal{V}} \sim O(1)$. While their corrections decay with the distance, at intermediate distances they have to be taken into account. This is true also for corrections to energy levels which we proceed to discuss.

## 3.2 Corrections to the spectrum on the sphere

In this paper we are rather interested in corrections to CFT spectrum on the sphere and not in the correlators. For this purpose, we consider the EFT of the form (13) but on the cylinder $S_R^{d-1} \times \mathbb{R}$:

$$S_{\text{CFT}}|_{\text{cyl}} + \sum_{\mathcal{V}} G_{\mathcal{V}} \int d\tau \int_{|n|=R} \mathcal{V}_{\text{cyl}}(\tau, n). \qquad (16)$$

Here $\mathcal{V}_{\text{cyl}}$ are the local CFT operators on the cylinder: $\tau \in \mathbb{R}$ is the imaginary time, $n \in S_R^{d-1}$ the coordinates on the sphere, and $\int_{|n|=R}$ is the integral with uniform measure over this sphere. The correlators of $\mathcal{V}_{\text{cyl}}$ are obtained from the CFT correlators in flat space via the Weyl transform. For the effective description to make sense, we assume that the radius of the sphere is much larger than the short distance cutoff: $R \gg 1$. We will also assume that $R \ll \xi$. We will see that corrections to CFT energy levels will then be small.

To evaluate the energy levels, it's instructive to pass to the Hamiltonian formalism. The Hamiltonian corresponding to the action (16) is obtained as the sum of the CFT Hamiltonian on $S_R^{d-1}$ and of the perturbing term, which is the $\tau$-integrand in (16) evaluated at $\tau = 0$ [56]:

$$H(R) = H_{\text{CFT}}(R) + \sum_{\mathcal{V}} G_{\mathcal{V}} \int_{|n|=R} \mathcal{V}_{\text{cyl}}(0, n). \qquad (17)$$

Here $\mathcal{V}_{\text{cyl}}$ is the same CFT operator as in (16).[9] The perturbation is assumed to be spherically symmetric, as is appropriate for the description of the fuzzy sphere model which has exact spherical symmetry at the microscopic level.[10]

---

[8]For simplicity we do not discuss marginal couplings, but if they exist they also need to be small.

[9]From perturbative field theory point of view, it may look like operators containing time derivatives change when going to the Hamiltonian formalism, due to the Legendre transform. But this change only affects the operator's expression in terms of fundamental fields. The matrix elements and correlators are unchanged, so from the CFT point of view, it is the same operator and is denoted accordingly.

[10]If instead only a discrete subgroup $\Gamma$ of $SO(d)$ is preserved, the couplings $G_{\mathcal{O}}$ may have nontrivial dependence on the sphere coordinates, consistent with $\Gamma$-invariance, as in [29] where $\Gamma$ was the icosahedral subgroup of $SO(3)$.

To make the dependence on $R$ more manifest, let us rescale the sphere radius to 1 (the short distance cutoff now becomes $a = 1/R$). The CFT operators rescale as $\mathcal{V} \to R^{-\Delta_\mathcal{V}} \mathcal{V}$, and the Hamiltonian takes the form:

$$H(R) = \frac{1}{R}\tilde{H}, \qquad\qquad \tilde{H} := H_{\text{CFT}} + V, \qquad\qquad (18)$$

$$V = \sum_{\mathcal{V}} g_{\mathcal{V}} \int_{|n|=1} \mathcal{V}_{\text{cyl}}(0, n), \qquad g_{\mathcal{V}} = G_{\mathcal{V}} R^{d-\Delta_\mathcal{V}}. \qquad (19)$$

In what follows we discuss the eigenvalues of the reduced Hamiltonian $\tilde{H}$. Let's work in the basis of states $|\mathcal{O}\rangle$ associated with the local CFT operators. As discussed in Sec. 2.2, the CFT Hamiltonian is diagonal in this basis, with energies $E_{\mathcal{O}} = \Delta_{\mathcal{O}}$ (or $\Delta_{\mathcal{O}} + w$ in presence of the Weyl anomaly).

We could obtain nonperturbative results for the energy levels by diagonalizing the infinite matrix $H_{\text{CFT}} + V$, e.g. truncating to finite size and extrapolating, referred to as the Truncated Conformal Space Approach [56–58]. This would be needed for $R \gtrsim \xi$, when corrections to CFT energy levels are significant. However, since in this paper we work at $1 \ll R \ll \xi$, perturbation theory will be sufficient for our purposes. Indeed, in this case the couplings $g_{\mathcal{V}}$ in (19) are all small:

- For irrelevant operators ($\Delta_\mathcal{V} > d$) we have $G_\mathcal{V} \sim O(1)$, and since $R \gg 1$, the coupling $g_\mathcal{V}$ gets suppressed by $R^{d-\Delta_\mathcal{V}}$;

- For relevant operators we use the assumption $R \ll \xi$ and the relation for $\xi$ from Eq. (15).

To first order in perturbation theory, corrections to non-degenerate energy levels are given by the standard quantum mechanical formula:

$$\delta E_\Phi = \langle \Phi | V | \Phi \rangle \qquad\qquad\qquad (20)$$

$$= \sum_{\mathcal{V}} g_{\mathcal{V}} \int_{|n|=1} \langle \Phi | \mathcal{V}_{\text{cyl}}(0, n) | \Phi \rangle, \qquad (21)$$

assuming that the state is normalized $\langle \Phi | \Phi \rangle = 1$.

For higher-order corrections, it would be actually advantageous to go back to the Lagrangian description, reducing them to integrals of correlation functions with multiple operator insertions on the cylinder [32, 59].[11] Depending on the dimension of the perturbing operator, this may give UV divergences, regulated by the short-distance cutoff $a$, and requiring renormalization of the couplings [60, 61]. In this paper we will be mostly content with the first-order correction (21) for which these issues do not arise.

## 3.3 Matrix elements from the OPE coefficients

We now discuss how the matrix elements (21) are computed (see e.g. [29]). The basic idea is to represent the matrix element as the (integral of) correlator on the cylinder where the states $|\Phi\rangle$ and $\langle \Phi |$ are prepared by inserting the operators $\Phi_{\text{cyl}}$ in infinite past and $(\Phi_{\text{cyl}})^\dagger$ in infinite future. Then, one performs the Weyl transformation from the cylinder back to $\mathbb{R}^d$, which maps the cylinder correlator to a CFT three-point function in $\mathbb{R}^d$ with operators $\Phi$ and $\Phi^\dagger$ inserted at 0 and $\infty$. We will consider several cases of increasing degree of complexity.

---

[11]Such expressions compactly encode the infinite sums over intermediate states which one would obtain if one were to use higher-order Schroedinger-Rayleigh perturbation theory, also taking into account constraints from conformal symmetry on the matrix elements.

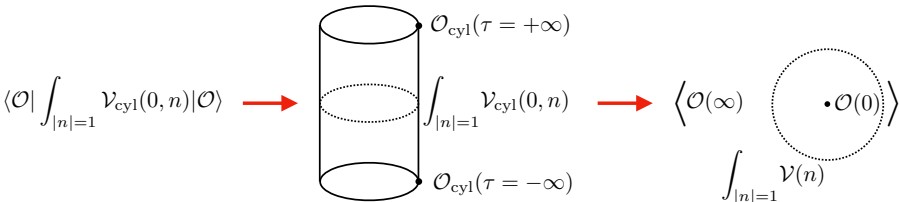

Figure 4: The matrix element (left), expressed as an integrated correlator on the cylinder where the bra and ket states are prepared by inserting operators $\mathcal{O}_{\text{cyl}}$ at infinite future and infinite past (center), and as an integrated CFT three-point function on $\mathbb{R}^d$ (right).

The simplest case arises when $\Phi = \mathcal{O}$ is a scalar primary, and the perturbing operator $\mathcal{V}$ is also a scalar primary. Then, after the Weyl transformation back to $\mathbb{R}^d$, the basic piece of matrix element (21) is expressed as (see Fig. 4 and explanations below):

$$\langle \mathcal{O}| \int_{|n|=1} \mathcal{V}_{\text{cyl}}(0,n)|\mathcal{O}\rangle = \int_{|n|=1} \langle \mathcal{O}(\infty)\mathcal{V}(n)\mathcal{O}(0)\rangle = \text{Vol}(S^{d-1}) f_{\mathcal{O}\mathcal{O}\mathcal{V}}, \qquad (22)$$

where $\mathcal{O}(\infty) := [\mathcal{O}(0)]^{\dagger} = \lim_{y\to\infty} |y|^{2\Delta_{\mathcal{O}}} \mathcal{O}(y)$. Here we used Polyakov's formula for the scalar primary three-point function [62], which implies $\langle \mathcal{O}(\infty)\mathcal{V}(n)\mathcal{O}(0)\rangle = f_{\mathcal{O}\mathcal{O}\mathcal{V}}$, where $f_{\mathcal{O}\mathcal{O}\mathcal{V}}$ is the OPE coefficient in the standard normalization.[12] Note that the result is spherically symmetric as expected. Note also that the state is unit normalized: $\langle \mathcal{O}|\mathcal{O}\rangle = \langle \mathcal{O}(\infty)\mathcal{O}(0)\rangle = 1$ working in the usual normalization of scalar primaries. The final result for the energy correction is given by (the sum over $\mathcal{V}$ is implied)

$$\delta E_{\mathcal{O}} = g_{\mathcal{V}} \text{Vol}(S^{d-1}) f_{\mathcal{O}\mathcal{O}\mathcal{V}}. \qquad (23)$$

Consider next the energy corrections for states which are *descendants* of a scalar primary, i.e. associated with a derivative of finite order of a scalar primary operator $\mathcal{O}$. These corrections are related to the corrections for the primary by some universal factors fixed by conformal invariance. There are two ways to recover these relative factors, either by using conformal algebra, or by mapping the matrix elements of the descendants to (integrals) of CFT correlation functions $\langle [\mathcal{O}(w)]^{\dagger} \mathcal{V}(n)\mathcal{O}(x)\rangle$ differentiated in $x$ and $w$ before taking the limits $x, w \to 0$. (Recall the observation in Sec. 2.2 that the reflection map is extended to descendants by linearity.) We also have to be careful to normalize the descendant states properly.

The descendants of $\mathcal{O}$ at level $k$ are degenerate of unperturbed energy $\Delta_{\mathcal{O}}+k$. They form a symmetric tensor product representation $\text{Sym}^k(\mathbf{d})$ where $\mathbf{d}$ is the fundamental of $SO(d)$. This representation is reducible for $k \geqslant 2$, and its irreducible components have different relative factors, so that the degeneracy is lifted by the perturbation. We describe here the result for $k = 1$, $d = 3$. The $k = 1$ descendants form a vector representation $\mathbf{3}$ of $SO(3)$ which is irreducible, and remains degenerate under the spherically symmetric perturbation. We may use (21) but we have to remember to normalize the states. The matrix element and the norm come out to be:

$$\langle \partial_{\mu}\mathcal{O}| \int_{|n|=1} \mathcal{V}_{\text{cyl}}(0,n)|\partial_{\nu}\mathcal{O}\rangle = 4\pi f_{\mathcal{O}\mathcal{O}\mathcal{V}} \left[1 + \frac{\Delta_{\mathcal{V}}(\Delta_{\mathcal{V}}-3)}{6\Delta_{\mathcal{O}}}\right] 2\Delta_{\mathcal{O}} \delta_{\mu\nu}, \qquad (24)$$

$$\langle \partial_{\mu}\mathcal{O}|\partial_{\nu}\mathcal{O}\rangle = 2\Delta_{\mathcal{O}} \delta_{\mu\nu}. \qquad (25)$$

---

[12]OPE coefficients of scalar primaries are normalized via $\mathcal{O}_1(x)\mathcal{O}_2(0) \supset f_{123}|x|^{\Delta_3-\Delta_1-\Delta_2}\mathcal{O}_3(0)$. This is the normalization used in all the literature. When one or more of the operators carry Lorentz indices, there is no commonly accepted convention. Our convention will be explained when needed.

Accounting for the normalization of the states, we obtain the energy correction

$$\delta E_{\partial\mathcal{O}} = 4\pi g_{\mathcal{V}} f_{\mathcal{O}\mathcal{O}\mathcal{V}} A(1,\mathbf{3}) = A(1,\mathbf{3})\delta E_{\mathcal{O}},$$
$$A(1,\mathbf{3}) = 1 + \frac{\Delta_{\mathcal{V}}(\Delta_{\mathcal{V}}-3)}{6\Delta_{\mathcal{O}}}, \tag{26}$$

where the notation $A(k,\rho)$ means that we are dealing with the correction for level $k$ descendants transforming in the irrep $\rho$ of SO(3).[13] Such relations between corrections for primaries and descendants mean that when CFT is perturbed, many energy levels will move in a correlated fashion. The derivation of (26) and analogous relations for descendants up to $k=4$ are discussed in App. B and in the notebook [34].

We will also need the case when the perturbing operator takes the form

$$\mathcal{V} = t_{\mu_1\ldots\mu_\ell}\mathcal{U}^{(\ell)}_{\mu_1\ldots\mu_\ell}, \tag{27}$$

a spin-$\ell$ primary operator $\mathcal{U}^{(\ell)}$ contracted with a coefficient tensor $t$ to get a rotationally invariant object.[14] The coefficient tensor $t$ in (27) must be built out of $\delta_{\mu\nu}$ and of the unit vector $n_\mu \in S^{d-1}$. The primaries being symmetric traceless tensors, the only nonvanishing contraction is

$$\mathcal{V}(n) = n_{\mu_1}\cdots n_{\mu_\ell}\mathcal{U}^{(\ell)}_{\mu_1\ldots\mu_\ell}(n). \tag{28}$$

The corresponding matrix elements is expressed via the integral of the CFT three-point function:

$$\int_{|n|=1}\langle\mathcal{O}(\infty)n_{\mu_1}\cdots n_{\mu_\ell}\mathcal{U}^{(\ell)}_{\mu_1\ldots\mu_\ell}(n)\mathcal{O}(0)\rangle = q_\ell \mathrm{Vol}(S^{d-1})f_{\mathcal{O}\mathcal{O}\mathcal{U}^{(\ell)}},$$
$$q_\ell = \frac{\ell!}{(d/2-1)_\ell 2^\ell}\times 2^{\ell/2}, \tag{29}$$

where $(x)_\ell$ is the Pochhammer symbol. To derive this equation one needs to use the form of the three-point function scalar-scalar-(spin $\ell$), see e.g. [37, Eq. (23)]. The correction to the energy of the primary state is given by an equation analogous to (23):

$$\delta E_{\mathcal{O}} = g_{\mathcal{V}}\mathrm{Vol}(S^{d-1})q_\ell f_{\mathcal{O}\mathcal{O}\mathcal{U}^{(\ell)}}. \tag{30}$$

In App. B.2 we also provide relative factors for corrections to descendant levels from perturbations with spin.

The above discussion was general, but in practice the important perturbing operator will be the lowest spin-4 primary $C$, which describes the leading Lorentz invariance breaking. On the other hand, the lowest spin-2 primary, which is the conserved stress tensor $T_{\mu\nu}$ does not give an interesting effect. In that case, the contraction (28) is the CFT Hamiltonian density in the radial quantization. The matrix element of the integral over the sphere is therefore the energy of the state. The energy correction is proportional to the energy itself. The effect of this correction is therefore to renormalize the speed of light. Since we will be fitting the speed of light, we don't have to include this correction separately.

A comment is in order concerning normalization of the OPE coefficients $f_{\mathcal{O}\mathcal{O}\mathcal{U}^{(\ell)}}$. In this paper we use the normalization as in [13], which corresponds to the OPE $\mathcal{O}_1(x)\mathcal{O}_2(0) \supset 2^{\ell/2}f_{\mathcal{O}_1\mathcal{O}_2\mathcal{U}^{(\ell)}}|x|^{\Delta_\mathcal{U}-\Delta_{\mathcal{O}_1}-\Delta_{\mathcal{O}_2}}x_{\mu_1}\cdots x_{\mu_\ell}\mathcal{U}^{(\ell)}_{\mu_1\ldots\mu_\ell}(0)$. In this normalization the

---

[13]Irreps of SO(3) will be denoted interchangeably by spin in parentheses, or multiplicity in boldface. E.g. spin-1 irrep will be denoted as $(\ell=1)$ or $\mathbf{3}$.

[14]As noted after (13) we do not have to consider total derivatives. This is obvious from the Lagrangian point of view. In the Hamiltonian formalism, total derivative perturbations have vanishing matrix elements at first order. In higher orders they are redundant—can be removed by a similarity transformation of the Hamiltonian leaving the spectrum invariant [29, footnote 7]. See [33, Eq. (3.9)] for an example of the latter.

three-point function is $2^{\ell/2}$ times the expression given in [37, Eq. (23)], which does not contain $\ell$-dependent factors. This explains the factor $2^{\ell/2}$ in (29). The rest of $q_\ell$ in (29) comes from subtracting traces in the three-point function.

Finally, it will be interesting to consider corrections to energies of states corresponding to some operators $\mathcal{O}$ with spin and their descendants. The two operators of interest in this paper are the stress tensor $T_{\mu\nu}$ and the spin-4 operator $C$. We will consider their shifts due to the coupling $g_\varepsilon$, written as

$$\delta E_{\mathcal{O}} = 4\pi g_\varepsilon f_{\mathcal{O}\mathcal{O}\varepsilon}^{\text{shift}}, \qquad \mathcal{O} = T, C. \tag{31}$$

The proportionality coefficients $f_{\mathcal{O}\mathcal{O}\varepsilon}^{\text{shift}}$ can be expressed in terms of OPE coefficients $\lambda_{\mathcal{O}\mathcal{O}\varepsilon}^i$, known from the conformal bootstrap. In the case at hand there are several conformally invariant tensor structures and several OPE coefficients. This computation is explained in App. B.4 and the resulting coefficients $f_{\mathcal{O}\mathcal{O}\varepsilon}^{\text{shift}}$ are given in Eqs. (B.34),(B.38). The energy shifts of descendants are then related to those of primaries by universal factors, worked out in App. B.3 for the stress tensor descendants up to level 2.

# 4   Warmup: CFT and CPT for a 1+1D Ising spin chain

In order to illustrate the power of CPT in combination with numerical energy spectra, we study a well known 1+1D lattice model in this section—the antiferromagnetic Ising model in a transverse and a longitudinal magnetic field—which features a line of Ising critical points in the magnetic fields plane. CPT results for the perturbed 2D Ising CFT were derived in two papers by Reinicke [32, 33] almost fourty years ago. Together these data and results allow to infer the couplings constants perturbing the CFT at and in the vicinity of the critical point for finite size systems. Furthermore we can analyze the finite-size scaling of the inferred couplings constants and we can confront them to their expected RG scaling behaviour.

We study the following 1D lattice Hamiltonian:

$$H_{\text{TFI}} = J \sum_i \sigma_i^z \sigma_{i+1}^z - h_z \sum_i \sigma_i^z - h_x \sum_i \sigma_i^x, \tag{32}$$

i.e. the antiferromagnetic (J>0) Ising model in a uniform longitudinal ($h_z$) and transverse field ($h_z$). At $h_z = 0$ this model is unitarily equivalent to the standard ferromagnetic transverse field Ising model, which can be solved exactly using a mapping to fermions [63], with a critical point at $h_x = 1$. For $h_z \in (-2, 2)$ the model has a line of critical Ising points at values of $h_x$ which depend on $h_z$ and are not known exactly, as the model loses its integrability away from $h_z = 0$. For illustration purposes we will focus on the value $h_z = 1$ only and explore the critical point and its vicinity by tuning $h_x$. The model has lattice translation symmetry and spatial reflection symmetry with respect to a site (or the center of a bond). For $h_z \neq 0$, there is no microscopic on-site $\mathbb{Z}_2$ symmetry. Instead, the global $\mathbb{Z}_2$ symmetry of the Ising CFT describing the critical point corresponds in the microscopic model to the translation by one lattice spacing. (It is the same $\mathbb{Z}_2$ which breaks spontaneously in the Néel phase of our model.)

As reviewed in Sec. 2.1, the spectrum of the 2D Ising CFT on a circle arranges into Virasoro multiplets built on top of the three Virasoro primaries $\mathbb{1}, \sigma, \varepsilon$. Here we will recover the low-lying part of this spectrum from ED of Hamiltonian (32) acting on the chain of $N$ spins with periodic boundary conditions. We focus on the 3+4+4 levels corresponding to CFT operators

$$\mathbb{1}, T, T\bar{T}, \qquad \varepsilon, \partial\varepsilon, \partial^2\varepsilon, \partial\bar{\partial}\varepsilon, \qquad \sigma, \partial\sigma, \partial^2\sigma, \partial\bar{\partial}\sigma, \tag{33}$$

which are easily identifiable in the ED spectrum by their scaling dimension, spin, and $\mathbb{Z}_2$, as there are no nearby degenerate states with the same quantum numbers. The ED spectrum

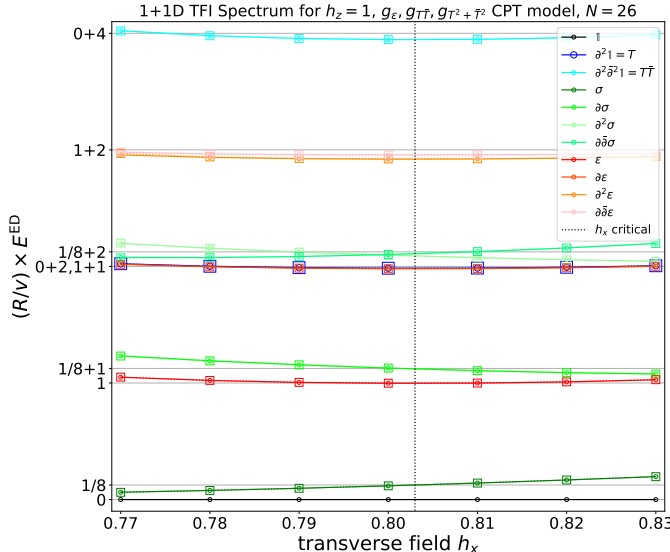

Figure 5: Comparison of the ED energy gaps with the 2D Ising CFT spectrum and with the 2D Ising spectrum corrected with the help of a CPT model. Circles and full colored lines: ED energy gaps for the Hamiltonian (32) for $N = 26$ rescaled by $R/v$ and plotted as a function of $h_x$ for $h_z = 1$. Horizontal gray lines: the exact 2D Ising CFT spectrum. Squares and dashed colored lines: the 2D Ising spectrum corrected by the Reinicke CPT model with couplings $g_\varepsilon$, $g_{T\widetilde{T}}$, $g_{T^2+\widetilde{T}^2}$ obtained by fit to $T, \sigma, \partial\sigma, \varepsilon, \partial\varepsilon$ levels. The vertical dashed line denotes the location of the critical point, determined from the equation $g_\varepsilon = 0$. The CPT model describes the ED energy gaps very well, despite the visible deviations of the ED energy gaps from the 2D Ising CFT values. The CPT and ED agreement is so good that the dashed lines are essentially invisible, hidden behind solid lines, for all the levels except $\partial^2\varepsilon$. Accordingly the circles are always inside the squares.

for these 11 states is shown in Fig. 5. We only study the energy gaps above the ground state assumed corresponding to the CFT operator $\mathbb{1}$. Using CPT fits described below we will see that the critical point for $h_z = 1$ is found at $h_x \approx 0.803$. Here we show the spectrum in its vicinity. We have rescaled the spectrum by multiplying the ED spectrum with $R/v$, where $v$ is the speed of light obtained from the CPT fit,[15] so that plotted energies would correspond to the values of scaling dimensions of the CFT in the absence of perturbations. A closer inspection of Fig. 5 reveals that the some of the energy levels drift quite a bit by tuning $h_x$ and also are not found at their expected CFT scaling dimension energy, therefore pointing to the presence of perturbing operators, which we are going to analyze in the following.

In what follows we will show that a much improved agreement between the ED and the CFT can be obtained by fitting the rescaled ED energy gaps to CPT predictions for the energy levels of 2D Ising CFT perturbed by relevant and irrelevant perturbations, obtained by Reinicke [32, 33] and reviewed in App. C.

As a first step we track the lowest two $\mathbb{Z}_2$-odd energy levels at momentum 0 and $2\pi/N = 1/R$,[16] which correspond to $\sigma$ and $\partial\sigma$. We then use the Reinicke formula to first order in $g_\varepsilon$ to infer the speed of light $v$ and the value of $g_\varepsilon$ for all values of $h_x$ shown.

---

[15]We use $R = Na/2\pi$, where $a = 1$ is the lattice spacing and serves as our unit of length, and $R$ is the radius of the circle.

[16]On the lattice these two levels have a momentum offset of $\pi/a$ because the magnetic order is antiferromagnetic in the model considered here. The $\mathbb{Z}_2$ quantum number is therefore encoded in whether the energy levels are found close to lattice momentum 0 ($\mathbb{Z}_2$-even) or $\pi/a$ ($\mathbb{Z}_2$-odd). Note that we consider even-length chains for simplicity.

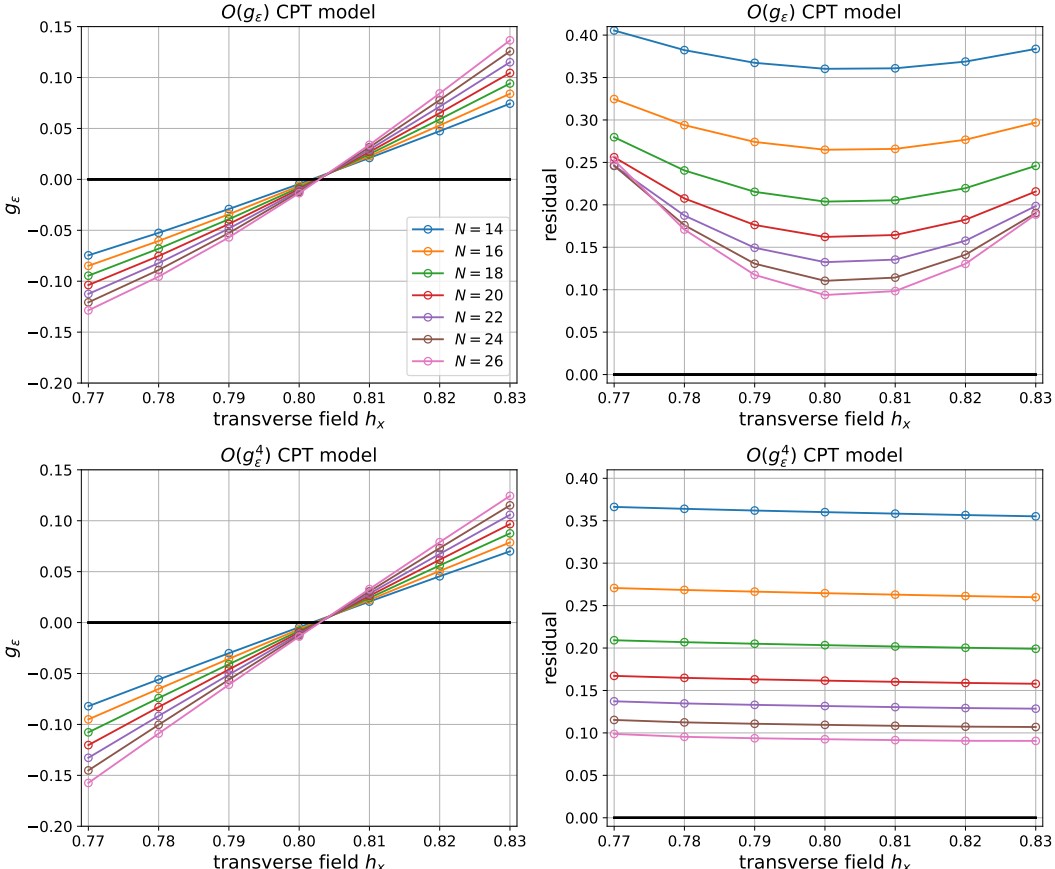

Figure 6: Analysis of the ED energy gaps with the help of the conformal perturbation theory model perturbing the 2D Ising CFT by the $\varepsilon$ operator. Top row: extracted $g_\varepsilon$ coupling from the first order in $g_\varepsilon$ model (left) and the residual differences between the CFT model at that order and the considered ED levels (right). Bottom row: extracted $g_\varepsilon$ coupling from the fourth order in $g_\varepsilon$ model (left) and the residual differences between the CFT model at that order and the considered ED levels (right).

The resulting $g_\varepsilon(h_x)$ is shown in the left panel of the upper row of Fig. 6 for system sizes $N$ ranging from 14 to 26 in steps of two. We see that the extracted $g_\varepsilon$ displays a zero crossing around $h_x \approx 0.803$, where it goes from a negative coupling (symmetry broken phase) to positive coupling (gapped, paramagnetic phase). In the right panel of the upper row we display the sum of the squared deviations from the expected CFT for the 10 excited ED levels considered. We observe that there is a minimum at the putative critical point, showing that the CPT model linear in $g_\varepsilon$ becomes inaccurate quickly when deviating from the critical point. Furthermore in the absence of other perturbing operators one would expect the deviations to vanish at the critical point. That we find a non-zero minimum points to the presence of other perturbing operators.

In the next step we include the full fourth order expression in $g_\varepsilon$ from the CPT formula, leading to a similar results for the extracted $g_\varepsilon(h_x)$ shown in the left panel, but now the residue is practically flat for the shown system sizes in the chosen $h_x$ window. So while the $h_x$ dependence of the spectrum is now well modeled within CPT, the remaining discrepancies between the finite size ED spectra and the CFT results still point to the presence of other operators perturbing the CFT spectrum.



**Figure 7:** Analysis of the ED energy gaps with the help of a more sophisticated conformal perturbation theory model perturbing the 2D Ising CFT by $\varepsilon, T\bar{T}, T^2 + \bar{T}^2$. Top row: extracted $g_\varepsilon$ coupling from the full CPT model (left) and the residual differences between the CFT model at that order and the considered ED levels (right). Middle row: extracted couplings $g_{T\bar{T}}$ (left) and $g_{T^2+\bar{T}^2}$ (right). Bottom row: the extracted couplings $g_{T\bar{T}}$ and $g_{T^2+\bar{T}^2}$ (left) and $g_\epsilon$ (right) at $h$ near $h_c$, plotted as a function of system size $N$. The dotted lines illustrate the scaling expected from the RG.

In the last step we include in the analysis the CPT predictions for the perturbations by two leading irrelevant parity-invariant operators of scaling dimensions 4: $T\bar{T}$ and $T^2 + \bar{T}^2$. We do so by fitting the CPT expressions for the ED energy levels $T, \sigma, \partial\sigma, \varepsilon, \partial\varepsilon$ with $\nu$, $g_\varepsilon$, $g_{T\bar{T}}$ and $g_{T^2+\bar{T}^2}$ as parameters. The results are shown in the first two rows of Fig. 7. While $g_\varepsilon(h_x)$ is almost unchanged compared to Fig. 6 on the scale of the plot, it is notable that the residual difference between the CPT and the ED levels dropped by more than a factor 20 by including the irrelevant operators in the fit. The extracted new coupling constants are shown in the middle row. For our chosen microscopic model the two couplings are both of negative sign, and of comparable magnitude.[17] While $T\bar{T}$ is the known leading scalar primary, which controls correction to scaling in the 2D Ising universality class, the presence of the spin-4 operator $T^2 + \bar{T}^2$ is due to presence of Lorentz symmetry breaking terms (e.g. a square lattice in 2D or a Hamiltonian formulation in 1+1D, leading to a space-time anisotropy). Furthermore the negative value of its coupling in our model is likely a band curvature effect which can be understood in the limit $h_z = 0$. Analogues of the two irrelevant operators will also play a role on our analysis of the fuzzy sphere geometry. In the bottom row we finally show the finite size behavior of the two extracted couplings $g_{T\bar{T}}$ and $g_{T^2+\bar{T}^2}$ at the critical point. Since these operators both have scaling dimensions 4, we would expect them to flow away with an exponent $2 - 4 = -2$ with increasing system size $N$, and the data shown in the bottom row nicely agrees with this expectation. Similarly, the value of $g_\varepsilon$ in Figs. 6(a) and 7(a) can be seen to increase approximately linearly with the system size away from criticality, in agreement with $2 - 1 = 1$.

As one can see in Fig. 5, this last CPT model describes the ED spectrum extremely well, so well that deviation is invisible by naked eye for all levels except for $\partial^2\varepsilon$.

# 5 Analysis of the 3D Ising fuzzy sphere spectrum

## 5.1 Hamiltonian

In this section we study the same microscopic Hamiltonian as in the pioneering work of Ref. [1]. The Hamiltonian is specified explicitly in App. D. For the present discussion the precise microscopics are not important. In essence the Hamiltonian has two effects or ingredients: i) the charged electrons carrying a spin[18] one-half degree of freedom move on the surface of the sphere under the effect of a perpendicular orbital magnetic field. The filling of the spherical Landau level is one electron per angular momentum orbital and under the effect of the interactions the electrons form a Mott insulating state, which means that the charge gap is finite, while the charges form an integer quantum Hall state with liquid-like charge correlations, i.e. they do not form a crystal. ii) The purpose of the interactions involving the spin degree of freedom of the electrons is to implement the competition between Ising interactions among spatially nearby electrons wanting to align the spin degrees of freedom ferromagnetically along the spin $\pm z$ direction on the one hand, and the transverse field term which strives to polarize the spins in the transverse, $x$ direction. At a suitably chosen value of the ratio of the transverse field to the other interaction parameters, a quantum phase transition takes place between the two phases, believed to be described by a 3D Ising CFT, and the results of Ref. [1] illustrate this beautifully.

---

[17]It is known that for our model at $h_z = 0$ the extracted coupling $g_{T\bar{T}}$ would be zero to the considered order, while $g_{T^2+\bar{T}^2}$ is also negative [33]. In the free fermion case $h_z = 0$ this is due to lattice induced curvature of the dispersion deviating from linearity.

[18]This spin should be thought of as pseudospin since it does not couple to the orbital magnetic field.

## 5.2   Spectrum at criticality: Observation of finite size effects

While Ref. [1] presented an impressive number of primaries and descendants of the 3D Ising CFT with rather high accuracy, it is curious that for example the reported scaling dimension of the lowest primary, the $\sigma$ field, while being relatively accurate for small electron numbers, is converging only slowly with increasing system size. This motivated us to perform a more detailed analysis of the finite size effects governing the spectrum at small electron numbers. Given that the numerical results are already quite close to the results from conformal bootstrap, it is plausible to assume that the remaining corrections can be described using the tools of conformal perturbation theory [29].

In Fig. 8 we show the low-lying spectrum of the fuzzy sphere transverse field Ising model for parameters $V_0 = 4.75$ and $h = 3.16$, the same values as used in most of Ref. [1]. This spectrum has been obtained using exact diagonalization on up to 18 electrons (c.f. App. E.1), while for selected low lying levels on up to 32 electrons we used an MPS implementation (c.f. App. E.2) to obtain the corresponding energies.

Ref. [1] used the known scaling dimension $\Delta_T = 3$ of the conserved stress energy tensor $T^{\mu\nu}$ to remove the speed of light from the spectrum. We will show below that from a CPT point of view the lowest primary $\sigma$ is actually least affected by perturbations other than the $\varepsilon$ field, while the effect of the many other possible perturbing fields on the stress energy tensor energy level are basically not known. Therefore we use the CPT framework to determine the speed of light $v$ and $g_\varepsilon$ from the lowest two energy gaps from the vacuum to $\sigma$ and to $\partial\sigma$ based on the known conformal bootstrap values of $\Delta_\sigma$ and $f_{\sigma\sigma\varepsilon}$ and their extension to $\partial\sigma$. The details of this procedure will be given in Sec. 5.4 below. We then rescale the spectrum to set $v/R$ to one (see Eqs. (2),(12)) and plot the resulting energy gaps, see Fig. 8. Since we are working at fixed microscopic parameters $V_0$ and $h$ there is a residual value of $g_\varepsilon(N)$ left, which we did not yet compensate for in Fig. 8 (because the effect of $g_\varepsilon$ is not known precisely for all the levels shown).

It is worthwhile to discuss a few features which become apparent when plotting the full $N$ dependence of the energy levels / scaling dimensions this way. Fig. 8 is organized in two rows containing the $\mathbb{Z}_2$ even (odd) sector in the top (bottom) row. The columns are organized according to spin $\ell = 0$ to 4 from left to right, while within a subplot the system size increases from right to left. CFT primaries are denoted as follows:

- $\varepsilon, \varepsilon'$ etc are the first, second etc $\mathbb{Z}_2$-even scalar primaries;

- $T^{\mu\nu}, T'^{\mu\nu}$ are the first and second $\mathbb{Z}_2$-even spin-2 primaries;

- $C^{\mu\nu\rho\sigma}, C'^{\mu\nu\rho\sigma}$ are the first and second $\mathbb{Z}_2$-even spin-4 primaries;

- $\sigma, \sigma'$ are the first and second $\mathbb{Z}_2$-odd scalar primaries;

- $\sigma^{\mu\nu}, \sigma^{\mu\nu\rho}, \sigma^{\mu\nu\rho\sigma}$ are the first $\mathbb{Z}_2$-odd primaries of spin 2,3,4.

We observe that the lowest lying levels $\sigma$ and its first two descendant families together with $\varepsilon$ and its first descendant show quite small finite size effects and a visible trend to converge to the expected CFT scaling dimensions when increasing $N$. Furthermore it seems that many primaries even higher up in the spectrum (such as e.g. $\varepsilon'$, $T^{\mu\nu}$, $\sigma'$, $\sigma^{\mu\nu}$, $\sigma^{\mu\nu\rho}$) exhibit rather small finite size effects. On the other hand, descendants show typically stronger and sometimes non-monotonous finite size effects especially as $\ell$ gets large, even for the $\sigma$ descendants. Furthermore, several (avoided) level crossing can be spotted higher up in the spectrum. A prominent example are the $\square\varepsilon$ and $\varepsilon'$ levels which repel each other in an avoided level crossing, and even more pronounced their descendants at $\ell = 1$. This will become even more clear when discussing our approach to extract certain OPE coefficients in Sec. 6, where certain wavefunction mixing effects at intermediate $N$ become apparent.

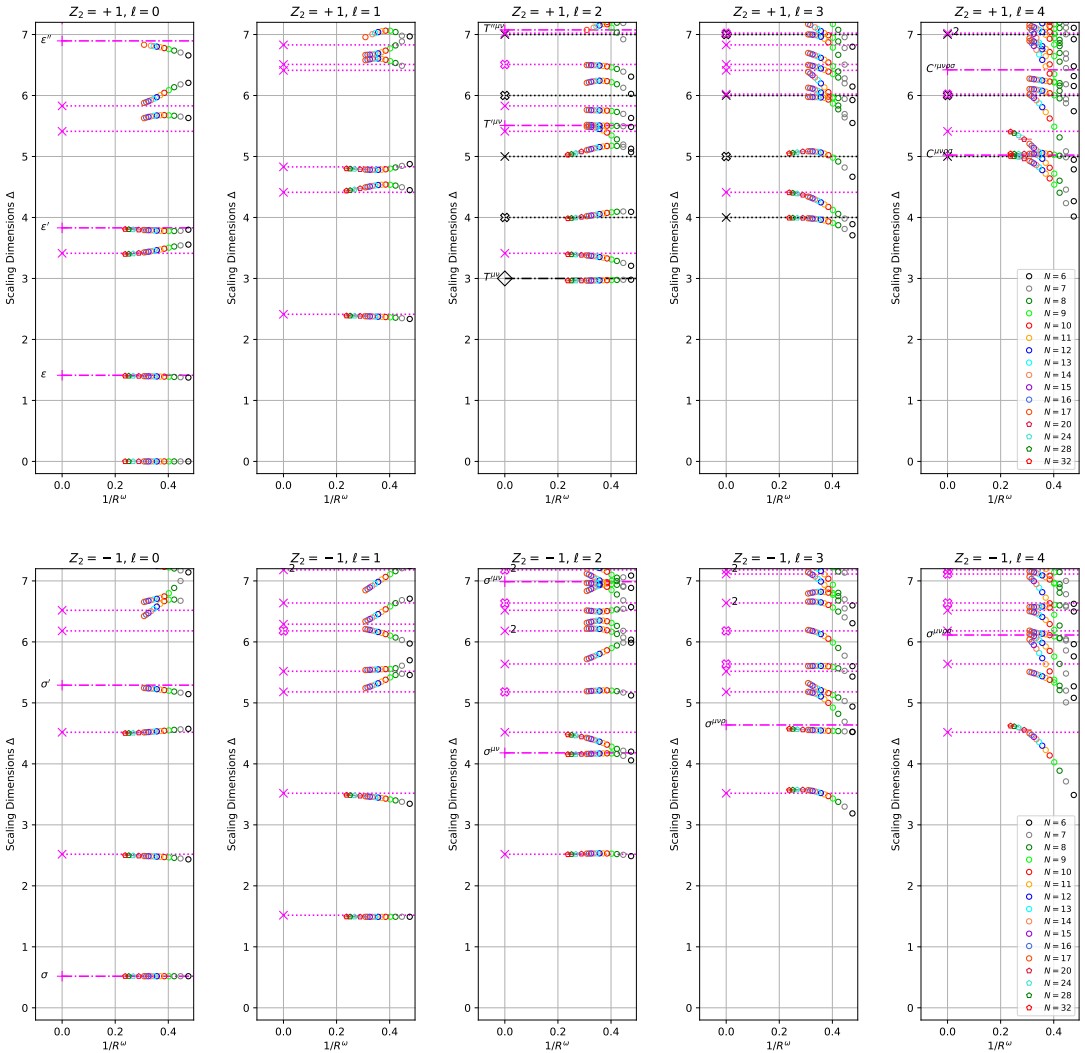

Figure 8: ED and MPS energy spectrum for the fuzzy sphere (2+1)D Ising Hamiltonian at $V_0 = 4.75$ and $h = 3.16$. Circle (pentagon) data points are obtained using ED (MPS). The speed of light required to rescale the spectrum has been obtained from the minimal CPT model discussed in Sec. 5.4. The horizontal lines with a symbol at their left end denote reference results (+: conformal bootstrap primary, ◇: exact primary, ×: parity-even descendants, empty cross: parity-odd descendants. Primaries are in addition labeled with a text tag, in the notation explained in the main text. When the conformal descendant multiplets are degenerate their multiplicity is indicated.

**Remark 5.1.** Looking closely at the $\mathbb{Z}_2 = -1$, $\ell = 4$ spectrum in Fig. 8, the state counting hints at another primary not much above $\sigma^{\mu\nu\rho\sigma}$. Available conformal bootstrap data contain no sign of such a primary. In this respect, it will be interesting to see the data coming from the extremal functional method from the recent conformal bootstrap study [45].

## 5.3 Conformal pertubation theory for the 3D Ising CFT on the fuzzy sphere

In the 2D Ising example discussed in Sec. 4 we obtained very good results based on the couplings $g_\varepsilon$, $g_{T\bar{T}}$ and $g_{T^2+\bar{T}^2}$. The next corrections would come from operators with scaling dimensions six, while we included primaries up to scaling dimension four.

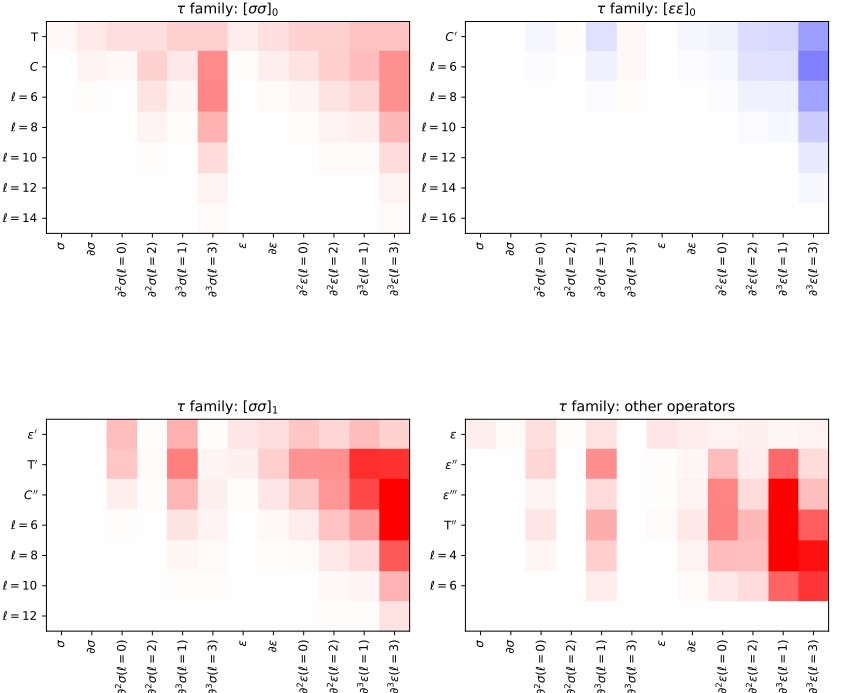

Figure 9: 3D Ising $\delta E$ pattern. The panels are organized according to families of fields in Ref. [13, Tables 3,4,5,7]. The $x$-axis labels the $\sigma$ and $\varepsilon$ fields and their lowest descendants up to level 3. The $y$-axis enumerates the perturbing fields in the corresponding families.

In order to devise a practical CPT scheme for the 3D Ising CFT on the fuzzy sphere it is useful to discuss the known operators which are allowed to contribute to the CPT. On the sphere, we can use as a perturbing operator any $\mathbb{Z}_2$-even primary $\mathcal{U} = \mathcal{U}^{(\ell)}$ with even spin $\ell$ and even spatial parity (in an appropriate rotationally invariant contraction if $\ell > 0$, see (28)). Several families of such primaries have been reported in Ref. [13] together with the OPE coefficients $f_{\sigma\sigma\mathcal{U}}$ and $f_{\varepsilon\varepsilon\mathcal{U}}$.

To first order, shifts of the $\sigma$ and $\varepsilon$ primary energy levels corrections are given by Eq. (30), and for their descendants by the same equation times the relative correction factors, which for the first few descendant levels were computed [29,34] (see App. B). The precise values of the couplings $g_{\mathcal{V}}$ appearing in this equation are unknown (they have to be determined from a fit), but their order of magnitude can be estimated by dimensional analysis as

$$g_{\mathcal{V}} \sim 1/R^{\Delta_{\mathcal{V}}-3}, \tag{34}$$

reflecting the fact that the couplings of higher-dimension operators should be suppressed, as discussed in Sec. 3.2. The sphere radius $R \propto \sqrt{N}$ with an unknown proportionality coefficient. In the plot below we ignore this expected dependence on $R$ and put $g_{\mathcal{V}} = 1$.

In this setup, we plot in Fig. 9 the expected shifts of the $\sigma$ and $\varepsilon$ primaries and the descendants energy levels from perturbations associated with operators from [13]. The figure is organized into 4 blocks which are grouped as in Ref. [13]. In each subplot the vertical axis labels perturbing fields in the family. On the horizontal axis we arrange the first few energy

levels in the $\sigma$ and $\varepsilon$ family. The intensity of the colored boxes (red positive, blue negative) displays the size of the expected correction. The plot reveals quite some structure, implying that not all energy levels are affected alike across energy levels and perturbing fields.

For example, one observes that the $\sigma$ energy level is basically only affected by the presence of a $g_\varepsilon$ couplings, while all the other shown perturbations have an insignificant effect. (As discussed previously the coupling to the stress energy tensor $T$ can be absorbed in the effective speed of light, for all states.) The first descendant of $\sigma$ is only affected by $g_\varepsilon$ and by $g_C$, where $C$ is the lowest spin-4 $\mathbb{Z}_2$-even primary. The $C$ perturbation is analogous to the perturbation $T^2 + \bar{T}^2$ discussed in Sec. 4 in the 2D case. Note that the leading $\mathbb{Z}_2$-even irrelevant scalar primary $\varepsilon'$ has little effect $\sigma$ and $\partial \sigma$, however in contrast it more significantly affects $\partial^2_{(\ell=0)}\sigma$, $\partial^3_{(\ell=1)}\sigma$ and the $\varepsilon$ family. Another observation is that descendants at low spin tend to be mostly affected by low-spin perturbations, while high-spin descendants couple to a larger variety of fields.

The complex structure of expected shifts in Fig. 9 shows that a fit of a large collection of perturbed energy levels using a small number of perturbing couplings may be delicate to justify. Since we will be operating at a relatively small number of electrons $N$, i.e. at small $R$, one could worry about the need to infer potentially many couplings from fits. Another difficulty is that most OPE coefficients beyond the ones already discussed are not available from the conformal bootstrap, so that we can't predict the corrections on other energy levels beyond $\sigma$ and $\varepsilon$ levels and their lowest few descendants. Finally, it so happens that some states belonging to different conformal multiplets have numerically close energy levels, raising the question about the importance of off-diagonal matrix elements and the associated level-mixing effects.

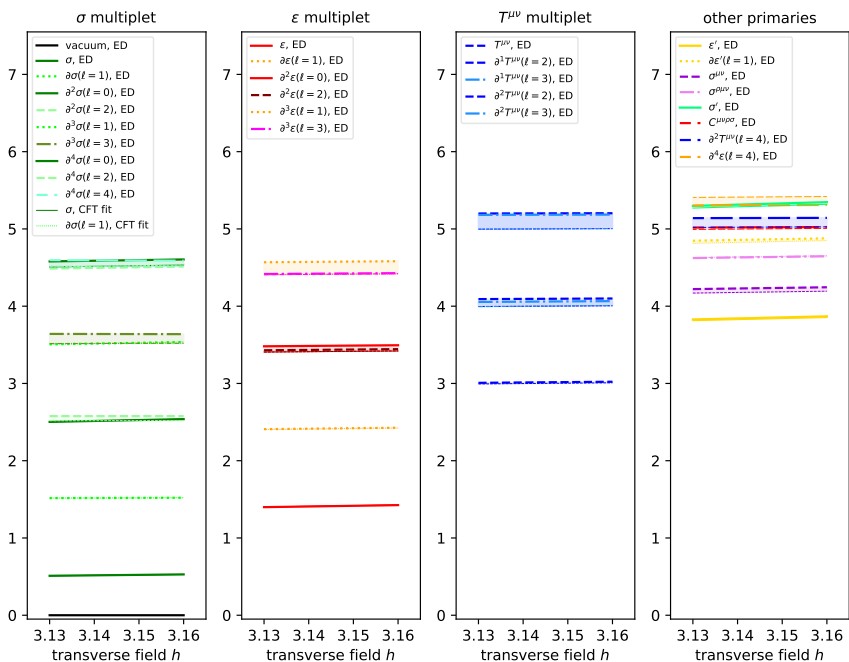

Figure 10: Dependence of ED spectrum when varying the microscopic $h$ coupling at constant $V_0 = 4.75$. Spectrum rescaled to remove the effect of the speed of light $v(h)$ by fitting to $\sigma$ and $\partial \sigma$. Note that many (but not all shown) higher lying ED levels are quite well accounted for by the minimal $g_\varepsilon$ CPT model discussed in Sec. 5.4.

Below we will be optimistic and will use first-order CPT with at most three couplings: the relevant and the two leading irrelevant perturbations $g_\varepsilon$, $g_{\varepsilon'}$, $g_C$.[19] We will see that this procedure works already rather well, although some small discrepancies do remain. Because of these discrepancies and due to the above-mentioned difficulties, we are currently not able to extend the procedure and obtain reliable fits for further CPT couplings. The level-mixing effects will be discussed in detail in Sec. 7.

Our focus will be to infer how CPT couplings $g_\varepsilon$, $g_{\varepsilon'}$, $g_C$ vary with $h$ and $V_0$. We will see that changing $h$ is directly linked to controlling $g_\varepsilon$, as expected. On the other hand $V_0$ controls $g_{\varepsilon'}$ and $g_C$. We will see that the fine tuned value of $V_0 = 4.75$ [1] corresponds to small values of $g_{\varepsilon'}$ and $g_C$. Moving $V_0$ away from $V_0 = 4.75$, both $g_{\varepsilon'}$ and $g_C$ start to grow with opposite signs.

## 5.4 Minimal CPT model: Fitting $v/R$ and $g_\varepsilon$

As for the 2D Ising model studied before, it is natural to expect that the transverse field $h$ is driving the transition from the $\mathbb{Z}_2$ ordered phase at small $h$ to the paramagnetic, massive phase at large $h$. We therefore expect that $h$ controls the coupling $g_\varepsilon$ which affects the finite volume CFT spectrum away from the critical point. We think of the term $(h - h_c) \int_{S^2} d\Omega\, \sigma^x(\Omega)$ in a microscopic Hamiltonian sense as having representation in the CFT sense in the immediate vicinity of the critical point as $g_{\mathbb{1}}\mathbb{1} + g_\varepsilon \varepsilon + g_{\mu\nu} T^{\mu\nu} + \dots$. We expect that coefficients for the left out operators will be small. Of those retained only $g_\varepsilon$ matters, as the other two ($g_{\mathbb{1}}$ and $g_{\mu\nu}$) can be absorbed into an overall energy shift and a correction to the speed of light.

In Fig. 10 we plot the low-energy spectrum for $N = 16$ electrons at fixed $V_0 = 4.75$ as a function of $h$. For each value of $h$ we use the energy gaps to $\sigma$ and $\partial\sigma$, identified as the first two $\mathbb{Z}_2$-odd ED levels $E_0^{\mathrm{ED}}(-, 0)$ and $E_0^{\mathrm{ED}}(-, 1)$, where $E_i^{\mathrm{ED}}(p, \ell)$ denotes the $i$th ED energy level in the $\mathbb{Z}_2$ parity sector $p$ at spin $\ell$, to determine the speed of light $v$ and the coupling $g_\varepsilon$. We start from the CPT formulas:

$$E_0^{\mathrm{ED}}(-, 0) - E_0^{\mathrm{ED}}(+, 0) = \frac{v}{R}\left(\Delta_\sigma + 4\pi \times g_\varepsilon \times f_{\sigma\sigma\varepsilon}\right), \tag{35}$$

and

$$E_0^{\mathrm{ED}}(-, 1) - E_0^{\mathrm{ED}}(+, 0) = \frac{v}{R}\left(\Delta_\sigma + 1 + 4\pi \times g_\varepsilon \times f_{\sigma\sigma\varepsilon} \times A(1, \mathbf{3})\right). \tag{36}$$

We can then easily solve these formulas for $v/R$ and $g_\varepsilon$ and therefore obtain estimates for $v/R$ and $g_\varepsilon$ as a function of the microscopic coupling $h$.[20] Note that this procedure relies on the known values of $\Delta_\sigma$ and $f_{\sigma\sigma\varepsilon}$ for the 3D Ising CFT. The spectrum in Fig. 10 is then shown after multiplying the ED gaps with $R/v$ (here we use $R = \sqrt{N}$).

Fig. 10 is organized for clarity of presentation into four panels, each of which contains a subset of the energy levels. The leftmost panel displays $\sigma$ and its descendants up to level 4. The second panel provides the same information for $\varepsilon$ and its descendants up to level 3. In the third panel we show the stress energy tensor and its level-one descendants. Finally in the last panel we show a few more primaries, both scalar and spinning. The thick lines present the ED data after removal of the speed of light. Thin lines of the same color show the CPT predictions $\Delta_{\mathcal{O}} + g_\varepsilon f_{\mathcal{O}\mathcal{O}\varepsilon}$, based on the extracted $g_\varepsilon$ values and the OPE coefficient $f_{\mathcal{O}\mathcal{O}\varepsilon}$ (times a relative correction factor in case of a descendant) which is either known from the conformal bootstrap or determined later in Sec. 6.

---

[19] In particular, we will omit the next irrelevant operator $T'$, of spin 2 and of dimension 5.50915(44) [13], i.e. somewhat above $C$. In the future it would be interesting to include $T'$ in the fit.

[20] While we do not show $v$ and $g_\varepsilon$ as a function of $h$ for this fit, we note that $v$ varies vary little in the shown range of $h$, while $g_\varepsilon(h)$ is very close to a plot in the second row of Fig. 11 for a more sophisticated fit. In particular $g_\varepsilon$ crosses zero at $h \approx 3.14$.

We see that the simple CPT model based on a determination of $g_\varepsilon$ by tracking $\sigma$ and $\partial\sigma$ as a function of $h$ and the knowledge of the OPE coefficients yields a very good agreement with surprisingly many other ED levels not included in the fit. A few levels are however not well captured by this simple CPT model. Such levels show in Fig. 10 as shaded bands, where the shading highlight the difference between the ED level and the CPT model prediction. So while the $h$ dependence is well captured by a model involving $g_\varepsilon$ to first order, a few ED levels differ from this model, and more sophisticated CPT models need to developed. We see that the remaining discrepancies (the widths of the shaded regions) are essentially independent of $h$ on the scale of this plot, suggesting that they may be corrected by couplings of irrelevant perturbations, which are approximately constant in a neighborhood of the critical point. The natural next step would be to add a coupling to the leading irrelevant scalar primary $\varepsilon'$ [29,31]. However, after having tried this we observed that this does not seem to improve the fit at $V_0 = 4.75$ too much, because many ED levels in principle affected by a finite $g_{\varepsilon'}$ are already well described without invoking $g_{\varepsilon'}$. So this correction is by itself unable to remedy the spectrum discrepancies without introducing discrepancies for energy levels where the agreement is already very good.

One particularly vexing issue concerns the avoided level crossing between $\Box\varepsilon$ and $\varepsilon'$ states. It's not obvious which coupling is responsible for this effect clearly visible in the data, see Sec. 7.

It's interesting to compare the situation of the fitting difficulties that we have here with the original work [29] which advocated using CPT for describing the ED spectra in the icosahedron model, and with the subsequent work [31] which applied CPT in the context of a fuzzy sphere model, where no such difficulties were noted. The main difference is that the criterion for declaring success was lower in [29,31], which dealt with the models which were not finetuned, so that the initial deviations from CFT were quite significant, and the dramatic improvement of the agreement after using the CPT was easier to see. Here on the contrary we are dealing with the model which was already carefully tuned in [1] by choosing $V_0 = 4.75$, to minimize deviations from CFT. In the future, it would be extremely important to isolate which other effects could explain the remaining discrepancies at $V_0 = 4.75$ and further improve the fit. One possibility is level mixing, which will be discussed in Sec. 7.

## 5.5 $V_0$ dependence of CPT couplings

The initial fuzzy sphere work for the 3D Ising CFT used a specific value of $V_0 = 4.75$ for most of their simulations [1]. In this section we discuss in what sense this particular choice of $V_0$ is fine-tuned to minimize finite size effects. We do this by analyzing ED data for two additional values of $V_0$, namely $V_0 = 2.5$ and $V_0 = 6$.

Based on the analysis of the structure of the effects of perturbing operators on energy levels in the $\sigma$ and $\varepsilon$ families presented in Sec. 5.3, we devise the following scheme: We track the energy levels corresponding to $\sigma$, $\varepsilon$, $\partial\sigma$ and $\partial^2_{(\ell=2)}\sigma$, the idea being that those levels are affected almost exclusively by $g_\varepsilon$, $g_{\varepsilon'}$ and $g_C$.[21] Using these four energy gaps we can therefore determine $v/R$, $g_\varepsilon$, $g_{\varepsilon'}$ and $g_C$ using the corresponding first order CPT corrections to the CFT energy levels. Note that this is not a fit, as we have as many parameters as equations.

In Fig. 11 we perform this analysis for all three values of $V_0$ and plot the obtained coupling constants in a certain $h$ window around the critical point, indicated by the zero crossing of $g_\varepsilon$ as a function of $h$. We are then interested in analyzing the residual couplings when $g_\varepsilon = 0$. For $V_0 = 2.5$ we see in the top row of Fig. 11 that the new coupling $g_{\varepsilon'}$ is quite large and positive, with a value of $4\pi \times g_{\varepsilon'} \approx 0.07$ for $N = 16$. Furthermore, $g_C$ is also nonzero, but with a negative value instead. We can also see a monotonic decrease of the magnitude of these two subleading coupling constants as we increase the system size $N$, as expected from

---

[21] We neglect here a smaller effect of the $[\sigma\sigma]_0$ family spin-6 operator on $\partial^2_{(\ell=2)}\sigma$.



Figure 11: By tracking the energy levels $\sigma$, $\varepsilon$, $\partial\sigma$ and $\partial^2_{(\ell=2)}\sigma$, we extract the couplings coefficients $g_\varepsilon$ (first column), $g_{\varepsilon'}$ (second column) and $g_C$ (third column), for several values of $V_0$ (first row: 2.5, second row 4.75 and third row 6.0) and varying transverse field $h$. The effective critical field corresponds to $g_\varepsilon = 0$. For $V_0 = 4.75$ selected in Ref. [1], the remaining $g_{\varepsilon'}$ is one order of magnitude lower than for the other two $V_0$s, explaining the small observed finite size effects. $g_C$ is also significantly smaller, though presents in all cases peculiar, non-decreasing finite-size effects. The bottom line shows the scaling of the different couplings $g_\mathcal{V}$ off-criticality ($h = 3$) as a function of the radius $R$ for the three $V_0$, compared to the expected behavior obtained from the CFT $R^{3-\Delta_\mathcal{V}}$. While in several cases, we observe a qualitative agreement, we observe several anomalies, e.g. $g_\varepsilon \propto R^{3-\Delta_{\varepsilon'}}$ for $V_0 = 2.5$, that we discuss in Sec. 5.5.1.

the renormalization group perspective. For the value $V_0 = 6$ we also observe sizable coupling constants in the lowest row of Fig. 11, however the signs of the coupling constants are now flipped compared to the $V_0 = 2.5$ case. The finite size behavior for $g_C$ is peculiar, as the extracted values seem to increase and then stagnate for the system sizes $N$ considered here. Finally in the middle row we show the application of this extended CPT model to the case $V_0 = 4.75$ considered before. The values of $g_{\varepsilon'}$ are now an order of magnitude smaller than for the other two considered values of $V_0$. This finding highlights that the chosen value $V_0 = 4.75$ of Ref. [1] seems to correspond to fine tuning $g_{\varepsilon'} \approx 0$, as conjectured in [29]. (The same conclusion was also reached in [22].) The remnant coupling $g_C$ at the critical point is roughly a factor two smaller in magnitude compared to the other two values of $V_0$, but is not behaving monotonously with increasing system size (the same observation also applies for the residual small value of $g_{\varepsilon'}$). So we see that despite the approximate vanishing of $g_{\varepsilon'}$, there are still some residual couplings or unaccounted effects at work, which deserve a better understanding. We will next discuss some possible approaches.

### 5.5.1 Comments on the $N$ dependence of CPT couplings

As mentioned several times the leading-order renormalization group intuition suggests that the CPT couplings should depend on the sphere radius $R = \sqrt{N}$ according to their scaling dimension: $g_{\mathcal{V}} \sim R^{3-\Delta_{\mathcal{V}}}$. Several examples of this behavior are tested in the bottom row of Fig. 11. In the first bottom panel we show that $g_\varepsilon$ coupling for $V_0 = 4.75$ measured at $h = 3$ (i.e. a bit off criticality) scales with $R$ according to scaling dimension of $\varepsilon$. In the second bottom panel we plot the $g_{\varepsilon'}$ coupling for $V_0 = 2.5$ and $V_0 = 6$ at $h = 3$, which show decrease with $R$, this time somewhat faster than the dimension of $\varepsilon'$ would suggest. The coupling $g_C$ at $V_0 = 2.5$ (the third bottom panel) is another case where the expectation also works more or less. But in other cases, which we don't highlight in the bottom row, the expectation does not work as nicely. First, for some $V_0$ the couplings $g_{\varepsilon'}$ and $g_C$ do not monotonically decrease with $R$. Second, for $V_0 = 2.5$ and $V_0 = 6$, the critical value of $h$ where $g_\varepsilon$ crosses zero drifts with $R$, indicating that $g_\varepsilon$ does not rescale homogeneously. Explaining these behaviors quantitatively is an interesting task beyond the scope of this paper. Still, we would like to discuss here possible reasons for these discrepancies, at a qualitative level, and focusing on the drift of $h_c$.

The first potential reason is that there are higher-order effects in the RG running. For example, at the second order the beta-function equations would have the form [55, p. 89]

$$\beta_i = R\frac{d}{dR}(4\pi g_i) = (3 - \Delta_i)(4\pi g_i) - \frac{1}{2}\sum_{jk} f_{ijk}(4\pi g_j)(4\pi g_k). \tag{37}$$

This equation is valid for scalar couplings. Spinful couplings such as $g_C$ can also be included in the analysis. In that case extra $O(1)$ prefactors will appear in the quadratic term. We will ignore these prefactors in the discussion below, since they don't modify the conclusion.

Could the second order term in the beta function explain the observed drift of $h_c$? The answer appears to be negative. To explain the drift, one needs an extra contribution to $4\pi g_\varepsilon$, compared to the overall rescaling by $(R_2/R_1)^{2-\Delta_\varepsilon}$, where $R_2 = \sqrt{16}$ and $R_1 = \sqrt{6}$. This extra needed contribution is about $-0.2$ for $V_0 = 2.5$ and about $0.1$ for $V_0 = 6$. The extra contribution from the quadratic term is an order of magnitude smaller than this. Moreover it has the same sign for $V_0 = 2.5$ and $V_0 = 6$, since all couplings flip sign.

The second possible reason for the drift in $h_c$ is as follows. When the microscopic theory (electrons in the magnetic field) is matched on the perturbed CFT at a length scale of the order of the magnetic length, the CPT couplings are expected to get corrections proportional to the curvature of the sphere $1/R^2 \sim 1/N$. One way to think about this is that there is a term in the effective action proportional to the Ricci scalar times the CFT operator. We were not paying

attention to such terms explicitly, since the Ricci scalar is anyway constant on the sphere of a given radius. But if one compares the theory on spheres of different radius, one becomes sensitive to such terms. In particular, we expect a $\delta \sim C/N$ contribution to the $g_\varepsilon$ coupling at the microscopic scale, where $C$ is an unknown $O(1)$ quantity of microscopic origin which may depend on $V_0$ and $h$. For a sphere of radius $R$, the corresponding extra contribution to $g_\varepsilon$ will be given by rescaling $\delta$ due to RG running:

$$R^{3-\Delta_\varepsilon} \delta \approx C N^{-0.2}, \tag{38}$$

i.e. essentially $N$ independent. This effect has thus the needed form to explain the observed drifts of $h_c$, provided that we assume that $C \sim -0.2$ for $V_0 = 2.5$, $C \sim 0$ for $V_0 = 4.75$, and $C \sim 0.1$ for $V_0 = 6$, with a weak dependence on $h$ in the studied range. This scenario requires a coincidence that the coefficient $C$ crosses zero at about the same $V_0$ as $g_{\varepsilon'}$.

The latter scenario can be in principle tested by going either to 1+1D or to the flat torus in 2+1D [64]. In both cases curvature is zero and the drift should be absent (up to higher order RG effects). Indeed, we have not observed a drift of $h_c$ in the 1+1D case studied in Sec. 4. We leave 2+1D torus tests to the future.

# 6 Extraction of OPE coefficients $f_{\mathcal{O}\mathcal{O}\varepsilon}$

The minimal CPT model developed in Section 5.4 relied only on the ED data for the $\sigma$ and $\partial\sigma$ levels. This is enough to translate the microscopic coupling $h$ into $g_\varepsilon(h)$ in the vicinity of the critical point. But in ED we have access to many more levels and their dependence on $h$ (and therefore on $g_\varepsilon$), as shown already in Fig. 10. From the structure of the leading order CPT we know that that the ED levels should exhibit a slope with respect to $g_\varepsilon$, of magnitude proportional to $f_{\mathcal{O}\mathcal{O}\varepsilon}$. We can turn this fact into a simple method to extract these OPE coefficients. We obtain ED energies in the vicinity of the critical point by tuning $h$. Using the minimal CPT model we convert the raw ED energies into the finite volume CFT spectral data as a function of $g_\varepsilon$. Calculating numerically the derivative of these rescaled energies with respect to $g_\varepsilon$ and evaluating it at $g_\varepsilon = 0$, we obtain directly numerical estimates for the OPE coefficients $f_{\mathcal{O}\mathcal{O}\varepsilon}$. Specifically, let us define

$$f_{\Phi\Phi\varepsilon}^{\text{ED}} = \frac{1}{4\pi} \frac{d\left[R/v \times \left(E_\Phi^{\text{ED}} - E_{\text{GS}}^{\text{ED}}\right)\right]}{dg_\varepsilon}\Bigg|_{g_\varepsilon=0}. \tag{39}$$

Then we expect,

$$f_{\Phi\Phi\varepsilon}^{\text{ED}} \approx \begin{cases} f_{\mathcal{O}\mathcal{O}\varepsilon}, & \Phi = \mathcal{O} \text{ scalar primary}, \\ A(k,\rho)f_{\mathcal{O}\mathcal{O}\varepsilon}, & \Phi \text{ level-}k \text{ descendant of a scalar primary } \mathcal{O}. \end{cases} \tag{40}$$

This procedure is approximate because couplings of irrelevant operators such as $g_{\varepsilon'}$ or $g_C$ also vary somewhat near the critical point. However, their variation is much slower than for $g_\varepsilon$, as can be seen from Fig. 11.

For $\mathcal{O}$ a spinning primary we expect

$$f_{\mathcal{O}\mathcal{O}\varepsilon}^{\text{ED}} \approx f_{\mathcal{O}\mathcal{O}\varepsilon}^{\text{shift}}. \tag{41}$$

Namely in this case there are in general many tensor structures in the conformal correlation function $\langle \mathcal{O}\mathcal{O}\varepsilon \rangle$, and the above procedure determines a particular combination of the corresponding OPE coefficients, which we denote $f_{\mathcal{O}\mathcal{O}\varepsilon}^{\text{shift}}$. In the case $\mathcal{O} = T, C$, we will be able

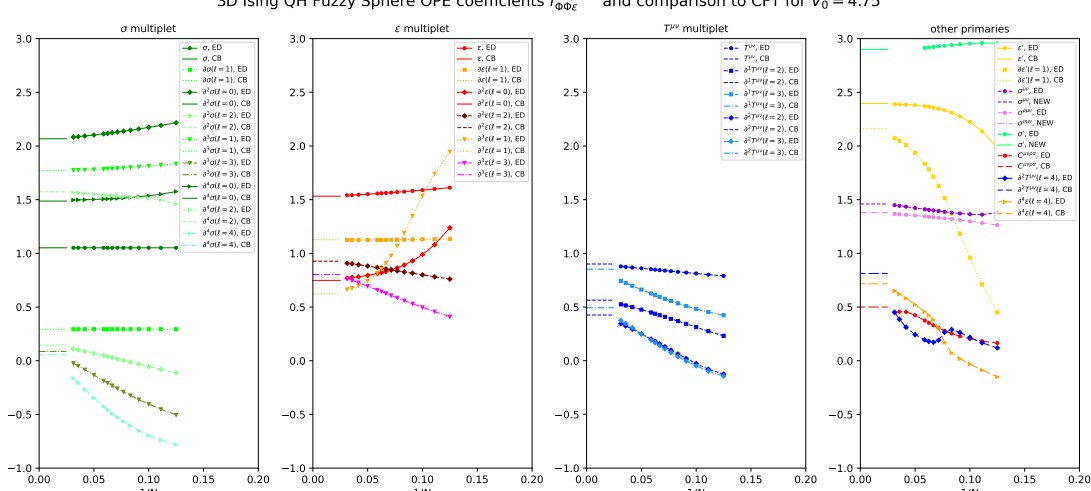

Figure 12: ED ($N = 8 \ldots 18$) and MPS ($N = 20, 24, 28, 32$) results for $f_{\Phi\Phi\varepsilon}^{\text{ED/MPS}}$ for the Ising model at $V_0 = 4.75$ and $h = 3.16$ and comparison to CFT prediction for $f_{\Phi\Phi\varepsilon}^{\text{shift}}$. Panels 1,2,3: $\sigma, \epsilon, T$ families. Panel 4: a few other primaries and descendants of $\varepsilon$, $T$ mixing with $C$. Conformal bootstrap data points have error bars, but they are not shown here for visual clarity of the figure.

to compare with the conformal bootstrap predictions for $f_{\mathcal{O}\mathcal{O}\varepsilon}^{\text{shift}}$ worked out in App. B.4. For the stress tensor descendants, the above procedure gives $\approx A_T(k, \rho) f_{TT\varepsilon}^{\text{shift}}$ where $A_T(k, \rho)$ is worked out up to $k = 2$ in App. B.3.

To use this procedure, we need to know how $g_\epsilon$ varies near the critical point. We use here the minimal fit which determines $g_\varepsilon$ and the speed of light from the first-order CPT for $\sigma$ and $\partial \sigma$ energy levels. The OPE coefficient $f_{\sigma\sigma\epsilon}$ used in this fit is assumed known and fixed to its CB value.[22]

This procedure delivers rather accurate and insightful data presented in Fig. 12. The plot is arranged into four columns, the left two being the $\sigma$ family (primary and descendants up to level four) and the $\varepsilon$ family (primary and descendants up to level three). The third columns contains the the stress energy tensor primary and descendants up to level 2. The rightmost column contains a selection of other low-lying primaries and some their descendants. Spin-4 descendants of $\varepsilon$ (level 2) and of the stress tensor (level 2) are also reported in this column for reasons that will become clear. Based on our prescription, the OPEs for $\sigma$ and $\partial \sigma$ are the ones we used to determine $g_\varepsilon$ in the first place, and hence identically agree with CFT predictions. All the other OPE estimates reported in this figure however are independent ED data. The short horizontal lines in the left of the corresponding panels are $f_{\Phi\Phi\varepsilon}^{\text{shift}}$ from other sources, either conformal bootstrap (CB) or prior fuzzy sphere work (see below).

In the $\sigma$ family we find that the low spin descendants of levels up to four exhibit quite small finite size effects and seem to approach their expected reference values for larger $N$. On the other hand the largest spin descendants of a given descendant level have small reference OPE coefficients and the numerical estimates exhibit larger finite size effects, starting off at numerical estimates even with the opposite sign. (A possible explanation is the approximation of neglecting variation of irrelevant couplings with $h$ may be worse off for higher level descendants.) Given the size of the exhibited finite size effects over the $N$ range covered here, it is plausible that the our numerical OPE estimates converge towards their expected values at larger $N$.

---

[22]One could alternatively say that this procedure determined ratios of other OPE coefficients divided by $f_{\sigma\sigma\epsilon}$.

In the $\varepsilon$ family we observe that the primary and the large spin descendants exhibit similar behavior as discussed for the $\sigma$ family. However we observe strikingly large finite size effects for the OPE coefficients of the levels labeled $\partial^2_{(\ell=0)}\varepsilon$ and $\partial^3_{(\ell=1)}\varepsilon$. Our interpretation of this finding is that there is level repulsion, i.e. an off-diagonal CPT matrix element from other perturbations beyond $g_\varepsilon$ which couples the two energetically nearby levels $\partial^2_{(\ell=0)}\varepsilon$ and $\varepsilon'$ and also $\partial^3_{(\ell=1)}\varepsilon$ and $\partial^1\varepsilon'$. There was first evidence for this effect already in an avoided level crossing visible in the energy spectrum itself as shown in Fig. 8, most clearly between $\partial^3_{(\ell=1)}\varepsilon$ and $\partial^1\varepsilon'$, and to a lesser extent between $\partial^2_{(\ell=0)}\varepsilon$ and $\varepsilon'$. See Sec. 7 below for a detailed discussion.

The third panel displays the OPE coefficients for the stress energy tensor $T^{\mu\nu}$ and some of its descendants. The OPE coefficient for the primary seems to converge smoothly towards the high precision rigorous CB result (B.34).[23] The descendants exhibit larger finite size effects but do not seem to signal a tension with the expected OPE coefficients for the descendants of the stress energy tensor.

In the last panel we show the extracted OPE coefficients for a few more primaries, such as $\varepsilon'$, $C$, $\sigma'$, $\sigma^{\mu\nu}$ and $\sigma^{\eta\mu\nu}$. While some of the considered levels have quite smooth and small finite size effects and will be discussed below, we see that $\varepsilon'$ and $\partial\varepsilon'$ show dramatically large effects. These are the respective partners of the levels discussed in the $\varepsilon$ family, which are coupled in the avoided level crossing scenario. And indeed they have the opposite behavior starting off from small apparent OPE values growing and then converging to OPE values of $2 \sim 2.5$, while the two concerned levels in the $\varepsilon$ families show the opposite trend. They start at seemingly large OPE coefficients for small $N$ and then drop rapidly and start to level off towards OPE coefficients around $0.6 \sim 0.7$. A natural interpretation is that at very small $N$ the order of two pairs of ED energy levels is inverted compared to the CFT expectation. Then at intermediate $N$ the two energy levels and their wave functions mix because of a large off-diagonal matrix element, while at large $N$ the expected CFT ordering is established and the mixing starts to fade out. We also observe a similar level reordering and accompanying avoided level crossings for three spin-4 states $C$, $\partial^2_{(\ell=4)}T_{\mu\nu}$ and $\partial^4_{(\ell=4)}\varepsilon$, however in this case the mixing matrix elements between the levels are much smaller, making the avoided level crossings apparent only in a much smaller $N$ range.

The other primaries not involved in a level repulsion scenario exhibit quite small finite size effects and allow us to report some new OPE coefficients $f^{\mathrm{shift}}_{\sigma'\sigma'\varepsilon} \approx 2.90$, $f^{\mathrm{shift}}_{\sigma^{\mu\nu}\sigma^{\mu\nu}\varepsilon} \approx 1.46$ and $f^{\mathrm{shift}}_{\sigma^{\kappa\mu\nu}\sigma^{\kappa\mu\nu}\varepsilon} \approx 1.38$. We are not attempting a finite size extrapolation, we just plot the bona fide results from our numerical procedure. One of these coefficients was previously reported in Ref. [14] as $f^{\mathrm{shift}}_{\sigma'\sigma'\varepsilon} \approx 2.98(13)$, measured in a different OPE extraction scheme in the fuzzy sphere setup, which yields exactly the same quantity in the $N \to \infty$ limit. We currently believe our result to be more accurate, as it does not rely on a similarly daunting extrapolation in $N$.

In conclusion, the OPE extraction method which we presented shows smaller finite size corrections than the previous method Ref. [14]. It is also relatively straightforward to implement. The main limitation of the method is that it is tailored to extracting OPE coefficients of the form $f_{\Phi\Phi\epsilon}$. We could also try to perturb the 3D Ising CFT by the $\sigma$ operator, which at the level of the microscopic model corresponds to weakly breaking the $\mathbb{Z}_2$ symmetry. However, the energy levels $E_\Phi$ will be corrected to second order in the $\sigma$ coupling, and these corrections would involve OPE coefficients $f_{\Phi\Psi\sigma}$ for all $\Psi$'s having the opposite $\mathbb{Z}_2$ quantum number from $\Phi$. So it's not clear if these scheme could lead to a method extracting the OPE coefficients involving the $\sigma$ operator.

---

[23]An alternative method from Ref. [14] required a bolder extrapolation in $N$ and produced a value somewhat below (B.34), see App. B.4. Our data supports (B.34) rather than the result of Ref. [14].

# 7 Level mixing effects

Looking at the energy level dependence in Fig. 8, one notices several pairs of levels with the same quantum numbers experiencing avoided level crossing as $N$ is varied. To explain this effect in the CPT framework requires to go beyond the diagonal matrix elements considered in the rest of this paper, and to include off-diagonal mixing. Let us discuss how this could in principle be achieved, taking two avoided level crossings as examples: (i) levels 3,4 with $\mathbb{Z}_2 = +$, $\ell = 0$, and (ii) levels 2,3 with $\mathbb{Z}_2 = +$, $\ell = 1$. The first pair is identified with the states $\square\varepsilon \equiv \partial^2_{(\ell=0)}\varepsilon$ and $\varepsilon'$, of nearby CFT dimensions $\Delta_\varepsilon + 2 \approx 3.41$ and $\Delta_{\varepsilon''} \approx 3.83$. The second pair are their first derivatives $\partial_\mu \square\varepsilon$ and $\partial_\mu \varepsilon'$. Avoided level crossing of these pairs of states is visible not only in their energy levels in Fig. 8, but also in the behavior of their apparent OPE coefficients $f^{\mathrm{ED}}_{\Phi\Phi\varepsilon}$, Fig. 12, as discussed in Sec. 6. We warn the readers that our analysis will be only partially successful in reproducing the numerical data, as we discuss at the end.

We will work in the approximation in which we include diagonal shifts to the involved energy levels $\Phi_1$ and $\Phi_2$, as well as their off-diagonal mixing, but neglect mixing with all other states. This appears reasonable because all other states are quite distant. In this approximation we compute the corrected energy levels by diagonalizing the $2 \times 2$ matrix

$$M = \begin{pmatrix} \Delta_1 & 0 \\ 0 & \Delta_2 \end{pmatrix} + 4\pi g_\mathcal{V} \begin{pmatrix} \delta_{11} & \delta_{12} \\ \delta_{12} & \delta_{22} \end{pmatrix}, \tag{42}$$

where

$$\delta_{ij} = \langle \Phi_i | \mathcal{V}(n) | \Phi_j \rangle, \tag{43}$$

and $\mathcal{V}$ is the perturbing operator, which we assume to be a scalar. The diagonal corrections $\delta_{ii}$ were discussed in Sec 3.3. The offdiagonal corrections necessitate a new computation.

**Case (i):** $\Phi_1 = \square\varepsilon$, $\Phi_2 = \varepsilon'$. The diagonal corrections are given by (Sec 3.3)

$$\delta_{11} = f_{\varepsilon\varepsilon\mathcal{V}} A_\varepsilon(2, \mathbf{1}), \qquad \delta_{22} = f_{\varepsilon'\varepsilon'\mathcal{V}}, \tag{44}$$

see [29, Eq. (C.10)] and [34] for $A_\varepsilon(2, \mathbf{1})$.[24] The offdiagonal correction is given by

$$\delta_{12} = \mathcal{N}^{-1/2} \langle \varepsilon' | \mathcal{V}(n) | \square\varepsilon \rangle \qquad (\mathcal{N} = \langle \square\varepsilon | \square\varepsilon \rangle) \tag{45}$$

$$= f_{\varepsilon\varepsilon'\mathcal{V}} B_\mathcal{V}, \tag{46}$$

where [34]

$$B_\mathcal{V} = \frac{\Delta_\mathcal{V}^2 - \Delta_\mathcal{V}(1 + 2r) + r + r^2}{\sqrt{12\Delta_\varepsilon(2\Delta_\varepsilon - 1)}} \qquad (r = \Delta_{\varepsilon'} - \Delta_\varepsilon). \tag{47}$$

**Case (ii):** $\Phi_1 = \partial\square\varepsilon$, $\Phi_2 = \partial\varepsilon'$. The corrections are now

$$\delta_{11} = f_{\varepsilon\varepsilon\mathcal{V}} A_\varepsilon(3, \mathbf{3}), \qquad \delta_{22} = f_{\varepsilon'\varepsilon'\mathcal{V}} A_{\varepsilon'}(1, \mathbf{3}), \qquad \delta_{12} = f_{\varepsilon\varepsilon'\mathcal{V}} \tilde{B}_\mathcal{V}, \tag{48}$$

where [34]

$$\tilde{B}_\mathcal{V} = \frac{(\Delta_\mathcal{V} - r)(\Delta_\mathcal{V} - r - 1)(\Delta_\mathcal{V}^2 - 3\Delta_\mathcal{V} + 10\Delta_\varepsilon - r^2 + 7)}{12\sqrt{5\Delta_\varepsilon(\Delta_\varepsilon + 1)(2\Delta_\varepsilon - 1)(\Delta_\varepsilon + r)}}. \tag{49}$$

The needed size of the off-diagonal coefficient $\delta_{12}$ can be estimated (up to a sign) as the distance of closest approach between the mixing states. From Fig. 8 we see that

$$4\pi g_\mathcal{V} \delta_{12} \sim \pm 0.25, \tag{50}$$

---

[24]In this section we will denote $A(k, \rho)$ as $A_\mathcal{O}(k, \rho)$, to emphasize whose descendants are involved.

is needed in both Cases (i) and (ii). Which perturbation $\mathcal{V}$ can achieve that? $\mathcal{V} = \varepsilon$ does not work because we are working at criticality and $g_\varepsilon$ is tuned to zero. $\mathcal{V} = \varepsilon'$ does not work either because we are working at $V_0 = 4.75$ and $g_{\varepsilon'}$ is very small, see Fig. 11, while the coefficients $B_{\varepsilon'} \approx 0.1$ and $\tilde{B}_{\varepsilon'} \approx 0.13$ are also small and suppress $\delta_{12}$ even further.

We next try $\mathcal{V} = \varepsilon''$, of dimension $\Delta_{\varepsilon''} = 6.8956(43)$ [13]. This gives larger coefficients $B_{\varepsilon''} \approx 2.79$, $\tilde{B}_{\varepsilon''} \approx 6.17$. Its needed OPE coefficients are:[25]

$$f_{\sigma\sigma\varepsilon''} = 0.0007338(31), \qquad f_{\varepsilon\varepsilon\varepsilon''} = 0.1279(17) \ [13], \tag{51}$$

$$f_{\varepsilon\varepsilon'\varepsilon''} \approx 2, \qquad f_{\varepsilon'\varepsilon'\varepsilon''} \approx 10 \ [66]. \tag{52}$$

The last two values [66] should be taken as order of magnitude estimate.

Using $f_{\varepsilon\varepsilon'\varepsilon''}$, we have

$$\delta_{12} \approx \begin{cases} 2 \times 2.79, & \text{Case (i)}, \\ 2 \times 6.17, & \text{Case (ii)}. \end{cases} \tag{53}$$

To reproduce (50), this translates into the needed coupling:

$$4\pi g_{\varepsilon''} \approx \pm \begin{cases} 0.045, & \text{Case (i)}, \\ 0.02, & \text{Case (ii)}. \end{cases} \tag{54}$$

The needed coupling is slightly smaller in Case (ii), consistently with the fact that the nearest approach is seen in Fig. 8 at a larger $N$ in this case. Indeed, we expect $g_{\varepsilon''}$ to decrease (in absolute value) in larger volumes due to RG running.

Because the OPE coefficients $f_{\sigma\sigma\varepsilon''}$ and $f_{\varepsilon\varepsilon\varepsilon''}$ are small, turning on the coupling $g_{\varepsilon''}$ of the size (54) should not significantly modify the fits of the $\sigma$, $\varepsilon$ energy levels and their descendants reported earlier.

Let us next discuss the diagonal elements $\delta_{11}$ and $\delta_{22}$ of the correction matrix in (42). This is where the discussed fit model will fail. For a consistent picture, we must have that the diagonal elements of $M$ cross,

$$\Delta_1 + \delta_{11} = \Delta_2 + \delta_{22}, \tag{55}$$

for $g_{\varepsilon''}$ which realizes the needed level splitting (50). Recall that $\Delta_1 < \Delta_2$. Furthermore, we see from Fig. 8 that the avoided crossing happens at a value $\Delta_c$ which is in between $\Delta_1$ and $\Delta_2$, in both Cases (i) and (ii). Thus we need $\delta_{11} > 0$, $\delta_{22} < 0$. Unfortunately, this is impossible in the discussed model where $\delta_{11}$ and $\delta_{22}$ have the same sign, both given by $g_{\varepsilon''}$ times a positive factor. Therefore, the described CPT model is unable to reproduce the precise shape of the numerical level repulsion curves under consideration. The best we could do is to choose $g_{\varepsilon''}$ negative, in which case the avoided level crossing would happen, with a roughly observed splitting, but near a $\Delta_c$ which is below both $\Delta_1$ and $\Delta_2$, and not in between them as seen in Fig. 8.

Resolving this discrepancy is an open problem which we leave to future work. One possibility is to include a further perturbing operator $\varepsilon'''$, of dimension 7.2535(51) [13]. Before going in this direction it would be necessary to ascertain the OPE coefficients of $\varepsilon''$ and $\varepsilon'''$ with $\varepsilon$ and $\varepsilon'$, improving on [66]. Another possibility is to try non-scalar $\mathcal{V}$. $\mathcal{V} = C$ would likely not work for the same reason as $\mathcal{V} = \varepsilon'$: because $g_C$ is already constrained to be small. On the other hand, $\mathcal{V} = T'$ is worth trying.

---

[25]The observed disparity in the size of the OPE coefficients is present already in the free theory. Namely we have $\sigma = \phi$, $\varepsilon = \frac{1}{\sqrt{2}}\phi^2$, $\varepsilon' = \frac{1}{\sqrt{4!}}\phi^4$, $\varepsilon'' = \frac{1}{\sqrt{6!}}\phi^6$, and $f_{\sigma\sigma\varepsilon''}^{\text{free}} = f_{\varepsilon\varepsilon\varepsilon''}^{\text{free}} = 0$, $f_{\varepsilon\varepsilon'\varepsilon''}^{\text{free}} = \sqrt{15} \approx 3.9$, $f_{\varepsilon'\varepsilon'\varepsilon''}^{\text{free}} = 8\sqrt{5} \approx 17.9$. See e.g. [65, App. A].

# 8 Conclusion and outlook

Numerical studies of continuous phase transitions in finite volume remain a valuable means to probe Conformal Field Theories (CFTs). For 1+1D models, the spatial manifold $S^1$ naturally facilitates comparisons with CFT due to radial quantization. In higher dimensions, $S^{d-1}$ achieves the same purpose, but using this in practice used to pose a challenge in achieving rotational invariance. The fuzzy sphere regulator introduced by Ref. [1] has proven to be a decisive advance in this direction. By employing a rotationally invariant Hamiltonian describing interacting electrons on $S^2$, this approach allowed for precise comparisons with the 3D Ising CFT data known from the conformal bootstrap, and led to insights into other observables not yet accessible by the bootstrap, and into other universality classes (see the references in the introduction). In this work, we extended the original study of [1] with an eye towards critically evaluating the method's efficacy and potential for systematic improvement.

Through the lens of Conformal Perturbation Theory (CPT), we demonstrated how to locate the critical point, determine the speed of light, and parametrize deviations between the fuzzy sphere model and the CFT. By analyzing the model away from the fine-tuned Haldane pseudopotential parameter $V_0 = 4.75$ reported in [1], we illustrated the correlation between the size of irrelevant CPT couplings and the finite-size effects. Our work highlights that while finite-size corrections remain significant for some energy levels, CPT provides a robust framework to mitigate these effects. In fact, CPT provides to some extent an alternative to tuning the model. Previously this was noted in [29] for the Transverse Field Ising Model on the icosahedron, and in [31] for an anionic fuzzy sphere model. In both cases the model was not tuned, but a meaningful comparison to CFT was nevertheless possible after (and only after) taking into account CPT corrections.

Additionally, we introduced a novel method for extracting OPE coefficients from the fuzzy sphere model. By analyzing the variation of energy levels with detuning from criticality, we provided a reliable scheme for extracting $f_{\mathcal{O}\mathcal{O}\varepsilon}$ coefficients, highlighting its effectiveness even in the presence of finite-size corrections. This analysis revealed interesting level mixing and avoided crossings, underscoring the complexities in interpreting numerical data at finite volumes.

In summary, our findings confirm the utility of the fuzzy sphere regulator and CPT for studying critical phenomena and extracting CFT data. While challenges remain, particularly with finite-size effects and level mixing, our methods offer a pathway for further refinement and increased understanding. Undoubtedly, the fuzzy sphere technique will continue to extend the range of phase transitions across various universality classes amenable to effective numerical study. We anticipate that the CPT technique will become a valuable and indispensable tool in increasing the efficacy of the fuzzy sphere method.

## Acknowledgments

AML acknowledges helpful discussions with G. Cuomo, A. Dey, Y.-C. He, J. Penedones, and R. Rattazzi. LH is thankful to Frédéric Mila for his support and discussions. SR is grateful to Joan Elias-Miró, Benoit Sirois, Bingxin Lao and Hugh Osborn for helpful discussions, and to the KITP and the EPFL for the hospitality. SR is especially grateful to Ning Su [66] and to David Poland, Valentina Prilepina and Petar Tadić [67] for communicating their unpublished results. AML and SR thank SCGP, Stony Brook for hospitality, where these results were first presented by AML at the workshop "Fuzzy Sphere Meets Bootstrap", November 6-8, 2023 [30].

**Funding information** LH was partially supported by the Swiss National Science Foundation Grant No. 212082. SR was supported in part by the Simons Foundation grant 733758 (Simons Bootstrap Collaboration). This research was supported in part by grant no. NSF PHY-2309135 to the Kavli Institute for Theoretical Physics (KITP).

# A  3D Ising CFT data

Primary operators of the 3D Ising CFT are characterized by their scaling dimension $\Delta$, spin $\ell$, and $\mathbb{Z}_2$ (spin flip parity) and $\mathbb{Z}_2^{O(3)}$ (spatial parity) quantum numbers. All primary operators below have $\mathbb{Z}_2^{O(3)} = 1$ unless mentioned otherwise.

The main source of information about the 3D Ising CFT spectrum is the conformal bootstrap studies [13] which used $\sigma, \varepsilon$ as the external operators. Ref. [13] reports many other operators appearing in the OPEs $\sigma \times \sigma$, $\varepsilon \times \varepsilon$, $\sigma \times \varepsilon$, whose dimensions and OPE coefficients were extracted via the extremal functional method [68], see Tables 2-7 in that work. We do not repeat those data from [13] here.

In a few cases we use conformal data from other conformal bootstrap studies, which are either more rigorous or more accurate than [13], or simply because [13] did not have access to that part of conformal data. This additional data is shown in Table 1. Notably we use Ref. [45] for the scaling dimensions of $\sigma, \varepsilon$ and their OPE coefficients among themselves, and [44, 66] for $\Delta_{\varepsilon'}$ and its OPE coefficients.

We will also use OPE coefficients $\lambda_{TT\varepsilon}$ and $\lambda_{CC\varepsilon}^i$ extracted from the conformal bootstrap in [45] and in [69, 70]. This will be discussed in App. B.4 where these OPE coefficients will be translated into matrix elements giving diagonal shifts of $T$ and $C$ states.

# B  CPT details

Here we will provide further details for Sec. 3. Some of these details have already been given in [29]. However some of the discussion of [29] is too complicated for our present purposes, since in [29] rotation invariance on $S^2$ was broken to a discrete subgroup (icosahedral symmetry), which led to additional splitting effects. We will try to make this appendix self-sufficient and will also connect the discussion to the accompanying Mathematica notebook [34] which should be consulted for the expressions and for the computations. For concreteness and since it's the main focus of this paper, the whole discussion will be limited to $d = 3$ but the same theory can be developed for any $d$.

Table 1: $\mathbb{Z}_2, \ell, \Delta$ and OPE coefficients of a few low-lying primary operators of the 3D Ising CFT which go beyond [13]. (The OPE coefficients are denoted by $\lambda_{ijk}$ in Ref. [44, 45] but the normalization is the same for scalar operators.) Boldface errors are rigorous, other errors are nonrigorous from the extremal functional method. Results from [66] have no error bar.

| $\mathcal{O}$ | $\mathbb{Z}_2$ | $\ell$ | $\Delta$ | $f_{\sigma\sigma\mathcal{O}}$ | $f_{\varepsilon\varepsilon\mathcal{O}}$ | $f_{\varepsilon'\varepsilon'\mathcal{O}}$ |
|---|---|---|---|---|---|---|
| $\sigma$ | $-$ | 0 | 0.518148806(**24**) [45] | 0 | 0 | 0 |
| $\varepsilon$ | $+$ | 0 | 1.41262528(**29**) [45] | 1.05185373(11) [45] | 1.53244304(58) [45] | 2.40 [66] |
| $\varepsilon'$ | $+$ | 0 | 3.82951(**61**) [44] | 0.05304(**16**) [44] | 1.5362(**12**) [44] | 7.68 [66] |
| $C$ | $+$ | 4 | 5.022665(28) [13] | 0.069076(43) [13] | 0.24792(20) [13] | 1.05 [66] |

We will need to evaluate matrix elements

$$\langle \Phi | \mathcal{V} | \Phi \rangle \,, \tag{B.1}$$

where $|\Phi\rangle$ is a state in the CFT Hilbert space associated with a primary operator $\mathcal{O}$ or its descendants (derivatives). We described in Sec. 2.2 how these matrix elements can be reduced to correlation functions in $\mathbb{R}^3$. The simplest examples were given in Sec. 3.

## B.1 $\mathcal{V}$ scalar primary, $\Phi$ descendants of a scalar primary

The calculation in this case starts from the CFT three point function

$$\langle \mathcal{O}(x_1) \mathcal{V}(x_2) \mathcal{O}(x_3) \rangle = f_{\mathcal{O}\mathcal{O}\mathcal{V}} F(x_1, x_2, x_3) \,, \tag{B.2}$$

$$F(x_1, x_2, x_3) = 1/(x_{12}^{\Delta_\mathcal{V}} x_{23}^{\Delta_\mathcal{V}} x_{13}^{2\Delta_\mathcal{O} - \Delta_\mathcal{V}}) \,, \quad x_{ij} := |x_i - x_j| \,. \tag{B.3}$$

Consider the case of second level descendants which is typical. The matrix element is computed as:

$$\langle \partial_{\nu_1} \partial_{\nu_2} \mathcal{O} | \mathcal{V}(n) | \partial_{\mu_1} \partial_{\mu_2} \mathcal{O} \rangle = \lim_{w, x \to 0} \partial_{w_{\nu_1}} \partial_{w_{\nu_2}} \partial_{x_{\mu_1}} \partial_{x_{\mu_2}} \langle [\mathcal{O}(w)]^\dagger \mathcal{V}(n) \mathcal{O}(x) \rangle \,, \tag{B.4}$$

where

$$\langle [\mathcal{O}(w)]^\dagger \mathcal{V}(n) \mathcal{O}(x) \rangle = |w|^{-2\Delta_\mathcal{O}} \langle \mathcal{O}(w_\mu / w^2) \mathcal{V}(n) \mathcal{O}(x) \rangle \,. \tag{B.5}$$

We then integrate over $n \in S^2$. The resulting matrix element

$$\langle \partial_{\nu_1} \partial_{\nu_2} \mathcal{O} | \int_{|n|=1} \mathcal{V}(n) | \partial_{\mu_1} \partial_{\mu_2} \mathcal{O} \rangle \,, \tag{B.6}$$

is a linear combination of tensors $\delta_{ab}\delta_{cd}$ where $abcd$ is a permutations of $\mu_1 \mu_2 \nu_1 \nu_2$, with coefficients which depend on $\Delta_\mathcal{O}$ and $\Delta_\mathcal{V}$. The scalar product

$$\langle \partial_{\nu_1} \partial_{\nu_2} \mathcal{O} | \partial_{\mu_1} \partial_{\mu_2} \mathcal{O} \rangle \,, \tag{B.7}$$

is obtained by setting $\mathcal{V} \to \mathbb{1}$ i.e. $\Delta_\mathcal{V} \to 0$.

Finally, we need to project both the matrix element and the scalar product on the irreducible irreps inside $\mathbf{3} \otimes \mathbf{3} = \mathbf{5} \oplus \mathbf{1}$. In the case at hand we use the projectors

$$P^{\mathbf{5}}_{a_1 a_2, b_1 b_2} = \frac{1}{2}(\delta_{a_1 b_1}\delta_{a_2 b_2} + \delta_{a_1 b_2}\delta_{a_2 b_1}) - \frac{1}{3}\delta_{a_1 a_2}\delta_{b_1 b_2} \,, \tag{B.8}$$

$$P^{\mathbf{1}}_{a_1 a_2, b_1 b_2} = \frac{1}{3}\delta_{a_1 a_2}\delta_{b_1 b_2} \,. \tag{B.9}$$

Expressing the projected matrix elements and the scalar product as

$$M_\rho P^\rho_{a_1 a_2, b_1 b_2} \,, \quad N_\rho P^\rho_{a_1 a_2, b_1 b_2} \,, \qquad (\rho = \mathbf{5}, \mathbf{1}) \,, \tag{B.10}$$

the energy correction to the states transforming in irrep $\rho$ is given by

$$\delta E_\rho = M_\rho / N_\rho \,. \tag{B.11}$$

We write it as $A(2, \rho)\delta E_\mathcal{O}$ where $\delta E_\mathcal{O}$ is the energy correction to the primary, Eq. (23).

In general, the relative factors $A(k, \rho)$ on level $k$ are functions of $\Delta_\mathcal{O}$ and $\Delta_\mathcal{V}$. The notebook [34] contains the expressions and the computations of all factors with $k \leqslant 4$:

$$A(1, \mathbf{3}) \,, \quad A(2, \mathbf{1}), A(2, \mathbf{5}) \,, \quad A(3, \mathbf{3}), A(3, \mathbf{7}) \,, \quad A(4, \mathbf{1}), A(4, \mathbf{5}), A(4, \mathbf{9}) \,. \tag{B.12}$$

The expressions for $k \leqslant 3$ also previously appeared in [29].

### B.1.1 Why Casimir eigenvalue?

Expressions for $A(k,\rho)$ in [29, 34] (see (26) for $A(1,\mathbf{3})$) show that $\Delta_{\mathcal{V}}$ enters through the conformal Casimir eigenvalue $C_{\mathcal{V}} = \Delta_{\mathcal{V}}(\Delta_{\mathcal{V}} - 3)$, and that furthermore

$$A(k,\rho) = 1 + O(C_{\mathcal{V}}).\tag{B.13}$$

In particular, for $\Delta_{\mathcal{V}} = 3$ (marginal) we get $A(n,\rho) = 1$, that is, all descendants get the same correction as the primary, and the splittings remain integer.

There can be understood via an alternative way of computing $A(k,\rho)$, using the conformal algebra generators acting in the Hilbert space of states on the sphere [46, 47]. This method is a bit harder to implement in Mathematica, but it does have some merits. We will now discuss this briefly for the first level descendants.

The generator $P_{\mu}$ produces the ket descendant states from the primary state:

$$|\partial_{\mu}\mathcal{O}\rangle = P_{\mu}|\mathcal{O}\rangle.\tag{B.14}$$

The special conformal generator $K_{\mu}$ is the conjugate of $P_{\mu}$ under the inversion of $\mathbb{R}^d$. We obtain the bra descendant states acting by it from the right:

$$\langle\partial_{\nu}\mathcal{O}| = \langle\mathcal{O}|K_{\nu}.\tag{B.15}$$

The matrix element we need to compute is now represented as

$$\langle\mathcal{O}|K_{\nu}\mathcal{V}(n)P_{\mu}|\mathcal{O}\rangle.\tag{B.16}$$

The idea now is to commute $K_{\nu}$ past $\mathcal{V}(n)$ towards the right, while $P_{\mu}$ towards the left, until the hit $\mathcal{O}$, at which point we use the primary state conditions [46, 47]

$$K_{\nu}|\mathcal{O}\rangle = 0, \qquad \langle\mathcal{O}|P_{\mu} = 0.\tag{B.17}$$

To perform this computation one can use the commutation relations of $K_{\mu}, P_{\nu}$ among themselves and with $\mathcal{V}(n)$:

$$[K_{\mu}, P_{\nu}] = 2\delta_{\mu\nu}D - 2M_{\mu\nu},\tag{B.18}$$

$$[K_{\nu}, \mathcal{V}(n)] = (2n_{\mu}(n\cdot\partial) - n^2\partial_{\nu} + 2\Delta_{\mathcal{V}}n_{\nu})\mathcal{V}(n), \qquad [P_{\mu}, \mathcal{V}(n)] = \partial_{\mu}\mathcal{V}(n).\tag{B.19}$$

Moreover, one can organize the computation so that the commutators with $\mathcal{V}(n)$ are never actually used explicitly. After integrating over $n \in S^2$, the matrix element is proportional to $\delta_{\mu\nu}$, and so it's enough to consider its contraction with $\delta_{\mu\nu}$, which we rewrite as follows:

$$\begin{aligned}\langle\mathcal{O}|K_{\mu}\mathcal{V}(n)P_{\mu}|\mathcal{O}\rangle &= \langle\mathcal{O}|[K_{\mu}, \mathcal{V}(n)]P_{\mu}|\mathcal{O}\rangle + \langle\mathcal{O}|\mathcal{V}(n)K_{\mu}P_{\mu}|\mathcal{O}\rangle\\ &= -\langle\mathcal{O}|[P_{\mu}, [K_{\mu}, \mathcal{V}(n)]]|\mathcal{O}\rangle + \langle\mathcal{O}|\mathcal{V}(n)[K_{\mu}, P_{\mu}]|\mathcal{O}\rangle,\end{aligned}\tag{B.20}$$

where in the passing to the second line we used (B.17).

Now, using the $[K, P]$ commutator (B.18), the second term in (B.20) is evaluated to

$$\langle\mathcal{O}|\mathcal{V}(n)2D\delta_{\mu\mu}|\mathcal{O}\rangle = 6\Delta_{\mathcal{O}}\langle\mathcal{O}|\mathcal{V}(n)|\mathcal{O}\rangle,\tag{B.21}$$

since $D|\mathcal{O}\rangle = \Delta_{\mathcal{O}}|\mathcal{O}\rangle$. It is this term which produces 1 in $A(1,\mathbf{3})$ given in (26).

For the first term in (B.20) we use the well-known expression for the quadratic Casimir of the conformal group (in general $d$ dimensions):

$$\mathrm{Cas}_2 = -P_{\mu}K_{\mu} + D(D - d) + \frac{1}{2}M_{\mu\nu}M_{\mu\nu}.\tag{B.22}$$

So we replace $-[P_\mu,[K_\mu,\mathcal{V}(n)]]$ by $[\text{Cas}_2,\mathcal{V}(n)]$ which gives the Casimir eigenvalue $C_\mathcal{V}=\Delta_\mathcal{V}(\Delta_\mathcal{V}-3)$ and explains the second term in $A(1,\mathbf{3})$ given in (26). Two other terms produced when replacing $-P_\mu K_\mu$ by $\text{Cas}_2$ vanish. The commutator with $M_{\mu\nu}M_{\mu\nu}$ vanishes as we are meant to integrate in $n$ and $\int_n \mathcal{V}(n)$ is rotationally invariant. The matrix element

$$\langle\mathcal{O}|[f(D),\mathcal{V}(n)]|\mathcal{O}\rangle\,, \qquad f(D)=D(D-3)\,, \tag{B.23}$$

vanishes because we are computing a diagonal matrix element and $f(D)$ gives the same eigenvalue $f(\Delta_\mathcal{O})$ when acting on $|\mathcal{O}\rangle$ and on $\langle\mathcal{O}|$.

It should be possible, and interesting, to generalize this argument to higher level descendants and explain in detail why $A(k,\rho)$ has the structure as in (B.13).

## B.2 $\ \mathcal{U}^{(\ell)}$ spinful primary, $\Phi$ descendants of a scalar primary

The relevant three-point function has the form [37, Eq. (23)]

$$\langle\mathcal{O}(x_1)\mathcal{U}_{\mu_1\dots\mu_\ell}(x_2)\mathcal{O}(x_3)\rangle=\tilde{f}_{\mathcal{O}\mathcal{O}\mathcal{U}}F(x_1,x_2,x_3)(Z_{\mu_1}\cdots Z_{\mu_\ell}-\text{traces})\,,$$
$$F(x_1,x_2,x_3)=1/(x_{12}^{\Delta_\mathcal{U}-\ell}x_{23}^{\Delta_\mathcal{U}-\ell}x_{13}^{2\Delta_\mathcal{O}-\Delta_\mathcal{U}+\ell})\,, \tag{B.24}$$
$$Z_\mu=\frac{x_{12}^\mu}{x_{12}^2}-\frac{x_{32}^\mu}{x_{32}^2}\,,$$

where $\tilde{f}_{\mathcal{O}\mathcal{O}\mathcal{U}}=2^{\ell/2}f_{\mathcal{O}\mathcal{O}\mathcal{U}}$.

We need to put $x_2=n$, $n^2=1$, and contract this correlator with $n_{\mu_1}\cdots n_{\mu_\ell}$, as in (28). We have

$$(Z_{\mu_1}\cdots Z_{\mu_\ell}-\text{traces})n_{\mu_1}\cdots n_{\mu_\ell}=K_\ell^{-1}|Z|^\ell P_\ell(Z.n/|Z|)_\ell\,, \tag{B.25}$$

where $P_\ell(x)$ is the Legendre polynomial, and

$$K_\ell=(1/2)_\ell 2^\ell/\ell!\,, \tag{B.26}$$

is the coefficient of the leading power $x^\ell$ in $P_\ell(x)$.[26] Then we have to set $x_1=w/w^2$, $x_3=v$ and expand the correlator around $v=w=0$ to order $n$, where $n$ is the descendant level one is interested in, as in the example (B.4) for scalar $\mathcal{V}$. In this limit we have $Z^2\to1$, $Z.n\to1$, and so the result will involve derivatives $(d/dx)^{k'}P_\ell(x)|_{x=1}$, $k'=0,\dots,k$. These derivatives can be easily computed from the Legendre differential equation; they are degree $k'$ polynomials in $\lambda=\ell(\ell+1)$.

Relative correction factors for descendants up to level $k=3$ are computed in the Mathematica notebook [34]. This reproduces Eqs. (C.34)-(C.37) from [29]. For reasons analogous to App. B.1.1, spin $\ell$ and dimension $\Delta_\mathcal{U}$ enter those results via the quadratic and quartic Casimir eigenvalues (see e.g. [71,72])

$$c_2=\Delta_\mathcal{U}(\Delta_\mathcal{U}-3)+\ell(\ell+1)\,, \qquad c_4=\ell(\ell+1)(\Delta_\mathcal{U}-1)(\Delta_\mathcal{U}-2)\,. \tag{B.27}$$

## B.3 $\ \mathcal{V}$ scalar primary, $\Phi$ descendant of the stress tensor

In this appendix we consider the case opposite to the previous one: perturbation $\mathcal{V}$ is a scalar primary, but the state we perturb is a primary with spin and its descendants. There are two issues to consider: 1) (this appendix) the relative shifts of descendants with respect to the primary; 2) (see App. B.4) normalization of matrix elements for the shifts of primaries in terms of OPE coefficients determined by the conformal bootstrap.

---

[26]In general $d$, $P_\ell(x)$ becomes the Gegenbauer polynomial $C_\ell^{(d/2-1)}(x)$, and $K_\ell=(d/2-1)_\ell 2^\ell/\ell!$.

We focus on the stress tensor $T$ and its descendants, up to the second level. Energy shifts of the descendants on level $k$ transforming in irrep $\rho$ are given by:

$$\delta E_T(k,\rho) = A_T(k,\rho)\delta E_T\,, \tag{B.28}$$

where $\delta E_T$ is the stress tensor shift. Taking into account the conservation, the first level descendants split as $\mathbf{7} + \mathbf{5}$, and the second level as $\mathbf{9} + \mathbf{7} + \mathbf{5}$. The relative factors are:

$$A_T(1,\mathbf{7}) = 1 + \frac{C_{\mathcal{V}}}{42}\,, \qquad A_T(1,\mathbf{5}) = 1 + \frac{C_{\mathcal{V}}}{6}\,, \qquad A_T(2,\mathbf{9}) = 1 + \frac{C_{\mathcal{V}}^2 + 134 C_{\mathcal{V}}}{3024}\,,$$

$$A_T(2,\mathbf{7}) = 1 + \frac{C_{\mathcal{V}}^2 + 70 C_{\mathcal{V}}}{336}\,, \qquad A_T(2,\mathbf{5}) = 1 + \frac{C_{\mathcal{V}}^2 + 22 C_{\mathcal{V}}}{84}\,. \tag{B.29}$$

To find these results it's best to use the algebraic method described in App. B.1.1. One considers the matrix element

$$\langle T_{\mu_1 \mu_2}(\infty)\mathcal{V}(n)T_{\nu_1 \nu_2}(0)\rangle\,. \tag{B.30}$$

It is constrained by scale invariance, rotational invariance, symmetry and tracelessness of the stress tensor, symmetry under the exchange of $0$ and $\infty$, and conservation of the stress tensor. It turns out that these constraints fix it uniquely in terms of $\Delta_{\mathcal{V}}$, up to an overall constant. Integrating the matrix element of the $n$-sphere gives the shift of the stress tensor level. The matrix elements for the descendants are obtained by acting on ket and bra states with $P$'s and $K$'s and commuting them past $\mathcal{V}(n)$, as described in App. B.1.1. One also needs to construct the projectors on various irreps into which the descendant levels split. These computations are carried out in [34].

## B.4 Matrix elements for shifts of $T$ and $C$ from $\mathcal{V} = \varepsilon$

When considering corrections to energy levels of primaries with spin such as $T$ or $C$ from the conformal perturbations such as $\varepsilon$, one faces the task of expression the matrix elements governing these shifts in terms of the conformal bootstrap OPE coefficients. This task is more complicated in this case, then say for shifts of scalar primaries, because the conformal three-point functions have several tensor structures.

Here we will perform the necessary translation for $T$ and $C$ shifts due to a scalar primary perturbation $\mathcal{V}$. Starting with $T$, the conformal three-point function $\langle TT\varepsilon\rangle$ is typically expanded in the so called box basis of tensor structures [73], nicely reviewed in [74, App. A]. Since we have $\ell_1 = \ell_2 = 2$, $\ell_3 = 0$, we have to put $n_{13} = n_{23} = 0$ in [74, Eq. (A.11)] and we have three tensor structures whose coefficients are denoted $\lambda_{TT\varepsilon}^i$, $0 \leqslant i = n_{12} \leqslant 2$.[27] In addition stress tensor conservation puts relations between these coefficients so that they can all be expressed in terms of $\lambda_{TT\varepsilon}^0$ ( [45, Eq. (2.2)]):

$$\lambda_{TT\varepsilon}^1 = \frac{4(\Delta_\varepsilon - 4)}{\Delta_\varepsilon + 2}\lambda_{TT\varepsilon}^0\,, \qquad \lambda_{TT\varepsilon}^2 = \frac{2(\Delta_\varepsilon^2 - 6\Delta_\varepsilon + 6)}{\Delta_\varepsilon(\Delta_\varepsilon + 2)}\lambda_{TT\varepsilon}^0\,. \tag{B.31}$$

The conformal bootstrap study [45], which included the stress tensor among external states, determined:

$$\lambda_{TT\varepsilon}^0 = 0.95331513(42)\,. \tag{B.32}$$

We will use this extremely accurate value. A less accurate prior conformal bootstrap determination was $0.958(7)$ [70], from the five-point bootstrap. To extract the energy level shift one

---

[27]For the considered case $\ell_2 = \ell_3$, the tensor structures in [Eq. (A.11)] [74] do not change sign under point permutations. Thus $\lambda_{TT\varepsilon}^i = \lambda_{T\varepsilon T}^i$.

has to take the limit of the three-point function [45, Eq. (2.2)] putting the stress tensors at 0 and $\infty$, and integrate the scalar over the sphere. This gives

$$\delta E_T = 4\pi g_\varepsilon f^{\text{shift}}_{TT\varepsilon}, \tag{B.33}$$

where [69, Eqs. (3.21),(3.22)]

$$\begin{aligned}
f^{\text{shift}}_{TT\varepsilon} &= \frac{2}{15}\lambda^0_{TT\varepsilon} - \frac{1}{3}\lambda^1_{TT\varepsilon} + \lambda^2_{TT\varepsilon} = \frac{4(\Delta_\varepsilon - 3)(\Delta_\varepsilon - 5)}{5\Delta_\varepsilon(\Delta_\varepsilon + 2)}\lambda^0_{TT\varepsilon} \\
&= 0.9008803(9),
\end{aligned} \tag{B.34}$$

where in the second line we substituted the conformal bootstrap values of $\lambda^0_{TT\varepsilon}$ and $\Delta_\varepsilon$.

We note that the quantity called the "$f_{TT\varepsilon}$ OPE coefficient" in the fuzzy bootstrap study [14] has exactly the same normalization as $f^{\text{shift}}_{TT\varepsilon}$ and should agree with it in the $N \to \infty$ limit. Yet they reported the value 0.8658(69), in discrepancy with CB. This is likely due to the difficulties of extrapolating to infinite volume in their setup. In Sec. 6 we present a different way of extracting $f^{\text{shift}}_{TT\varepsilon}$ from the fuzzy sphere data, which is less affected by finite size corrections and gives a value in agreement with CB.

The calculation for $C$ is similar. There are 5 tensor structures in the $\langle C\varepsilon C\rangle$ correlator, with their coefficients $\lambda^i_{C\varepsilon C}$, $0 \leqslant i \leqslant 4$.[27] The energy shift is given by

$$\delta E_C = 4\pi g_\varepsilon f^{\text{shift}}_{CC\varepsilon}, \tag{B.35}$$

where [34]

$$f^{\text{shift}}_{CC\varepsilon} = \frac{8}{315}\lambda^0_{C\varepsilon C} - \frac{2}{35}\lambda^1_{C\varepsilon C} + \frac{2}{15}\lambda^2_{C\varepsilon C} - \frac{1}{3}\lambda^3_{C\varepsilon C} + \lambda^4_{C\varepsilon C}. \tag{B.36}$$

The coefficients $\lambda^i_{C\varepsilon C}$ have been determined in a five-point bootstrap study [70] (improving on [69]). Using the last column in [70, Table 3]:

$$\{\lambda^i_{C\varepsilon C}\}_{i=0\ldots4} = \{-4(2), -10(2), 0.6(9), -0.4(2), -0.28(2)\}, \tag{B.37}$$

one obtains $f^{\text{shift}}_{CC\varepsilon} = 0.40(23)$.[28] Recently, the authors of [70] repeated their analysis including the high precision value of $\lambda^0_{TT\varepsilon}$ from [45] and extracting directly the linear combination (B.36) of interest to us. This gives a more accurate determination [67]:

$$f^{\text{shift}}_{CC\varepsilon} = 0.50(14), \tag{B.38}$$

which is the one we will use in Sec. 6.

# C  CPT results of Reinicke [32, 33]

Reinicke [32,33] developed CPT for the 2D Ising CFT. He considered Ising CFT states on $S^1$ of radius $R = 1$:

$$|h, \bar{h}\rangle = |X + r, \bar{X} + \bar{r}\rangle, \qquad r, \bar{r} \in \mathbb{Z}_{\geqslant 0}, \tag{C.1}$$

corresponding to local operators $\Phi_{h,\bar{h}}$ of conformal weights $h = X + r, \bar{h} = \bar{X} + \bar{r}$. The Virasoro descendants of $\mathbb{1}, \varepsilon, \sigma$ at level $r, \bar{r}$ where $(X, \bar{X}) \in \{(0, 0), (1/2, 1/2), (1/16, 1/16)\}$ are the weights of the Virasoro primary.[29] (Some of these descendant levels do not exist since they

---

[28]Since the correlation matrix of the $1\sigma$ errors on $\lambda^i_{C\varepsilon C}$ is not reported in [70], here we just combined all the errors with appropriate coefficients and $\pm$ signs chosen to maximize the overall error on $f^{\text{shift}}_{CC\varepsilon}$.

[29]In Reinicke's papers $X, \bar{X}$ are denoted $\Delta, \bar{\Delta}$.

are null, i.e., have zero norm.) Their unperturbed energies are $E_{h,\bar{h}} = h + \bar{h} - c/12$, and the energy gaps are

$$\mathcal{F}_{h,\bar{h}} = E_{h,\bar{h}} - E_{0,0} = h + \bar{h}. \tag{C.2}$$

He perturbed the CFT Hamiltonian $H_{\text{CFT}}$ with[30]

$$V = \sum_{\mathcal{V}} G_{\mathcal{V}} \int_{S^1} dx\, \mathcal{V}(0, x), \tag{C.3}$$

where $\mathcal{V}$ is one of the three operators $\varepsilon$, $T\bar{T}$ or $T^2 + \bar{T}^2$ (see footnote 2). He computed the energy gaps $\mathcal{F}_{h,\bar{h}}$ in perturbation theory to fourth order in $G_\varepsilon$ [32], and to first order in the other two couplings [33], for the cases when the unperturbed energy level $|h, \bar{h}\rangle$ has degeneracy exactly one, which happens for the 2D Ising CFT for

$$
\begin{aligned}
X = \bar{X} = 0: & \qquad r, \bar{r} \in 0, 2, 3, \\
X = \bar{X} = 1/2: & \qquad r, \bar{r} \in 0, 1, 2, 3, \\
X = \bar{X} = 1/16: & \qquad r, \bar{r} \in 0, 1, 2.
\end{aligned}
\tag{C.4}
$$

The $\varepsilon$ perturbation results are as follows ($G = 2\pi G_\varepsilon$):

$$\delta\mathcal{F}_{r,\bar{r}} = G^2[\alpha_{\mathbb{1}}(r) + \alpha_{\mathbb{1}}(\bar{r})] + G^4[\beta_{\mathbb{1}}(r) + \beta_{\mathbb{1}}(\bar{r})], \tag{C.5}$$

$$\delta\mathcal{F}_{\frac{1}{2}+r, \frac{1}{2}+\bar{r}} = G^2[\alpha_\varepsilon(r) + \alpha_\varepsilon(\bar{r})] + G^4[\beta_\varepsilon(r) + \beta_\varepsilon(\bar{r})], \tag{C.6}$$

$$\delta\mathcal{F}_{\frac{1}{16}+r, \frac{1}{16}+\bar{r}} = G\frac{1}{2}(2\delta_{r,0} - 1)(2\delta_{\bar{r},0} - 1) + G^2[\ln 2 + \alpha_\sigma(r) + \alpha_\sigma(\bar{r})]$$
$$+ G^4\left[\frac{3}{4}\zeta(3) + \beta_\sigma(r) + \beta_\sigma(\bar{r})\right], \tag{C.7}$$

where

$$\alpha_{\mathbb{1}}(r) = 1 + 1/(2r - 1), \qquad \beta_{\mathbb{1}}(r) = 1 + 1/(2r - 1)^3, \tag{C.8}$$

$$\alpha_\varepsilon(r) = 1/(2r + 1), \qquad \beta_\varepsilon(r) = 1/(2r + 1)^3, \tag{C.9}$$

$$\alpha_\sigma(r) = \frac{1}{2r}(1 - \delta_{r,0}), \qquad \beta_\varepsilon(r) = \frac{1}{8r^3}(1 - \delta_{r,0}). \tag{C.10}$$

Note that the order $g_\varepsilon$ corrections to the Virasoro primary levels $\varepsilon$ and $\sigma$ are in agreement with the general Eq. (23) and the OPE coefficients $f_{\varepsilon\varepsilon\varepsilon} = 0$, $f_{\sigma\sigma\varepsilon} = 1/2$.

The $T\bar{T}$ perturbation result is

$$\delta\mathcal{F}_{h,\bar{h}} = G_{T\bar{T}}[(h - c/24)(\bar{h} - c/24) - (c/24)^2], \tag{C.11}$$

while for the $T^2 + \bar{T}^2$ perturbation:

$$\delta\mathcal{F}_{X+r, \bar{X}+\bar{r}} = G_{T^2+\bar{T}^2}[A(X, r) + A(\bar{X}, \bar{r})], \tag{C.12}$$

where

$$A(X, r) = \begin{cases} \left(\frac{11}{30} + \frac{c}{12}\right)r(2r^2 - 3), & X = 0, \\ (X + r)\left[X - \frac{1}{6} - \frac{c}{12} + \frac{r(2X+r)(5X+1)}{(X+1)(2X+1)}\right], & X \neq 0. \end{cases} \tag{C.13}$$

Results (C.5)-(C.7) for the $\varepsilon$ perturbation are 2D Ising CFT specific. On the other hand, Eqs. (C.11) and (C.12) for $T\bar{T}$ and $T^2 + \bar{T}^2$ perturbations are valid not just in the 2D Ising but in any 2D CFT and for any nondegenerate energy level.[31]

---

[30]In Reinicke's papers the Hamiltonian is defined on the circle of radius $R = 1/2\pi$, while we consider $R = 1$ to simplify comparison to other parts of our paper. We transform his results to our conventions.

[31]In the 3D case, $\varepsilon'$ will be analogue of $T\bar{T}$ and $C$ will play the role of $T^2 + \bar{T}^2$. In that case the corrections to energy levels will involve the OPE coefficients which are theory dependent. But in 2D, matrix elements of $T\bar{T}$ and $T^2 + \bar{T}^2$ are universal functions of external state dimensions.

Next let us consider that the CFT is perturbed by the same perturbation $\sum G_{\mathcal{V}}\mathcal{V}$ but on the circle $S_R^1$ of radius $R$ different from 1. Rescaling the circle to $R = 1$ we obtain an overall rescaling $1/R$ of the spectrum, accompanied by the rescaling of the coupling constants

$$G_{\mathcal{V}} \to g_{\mathcal{V}} = G_{\mathcal{V}}/R^{\Delta_{\mathcal{V}}-2}.\tag{C.14}$$

Using the above formulas, we obtain that the energy gaps on $S_R^1$ will be given by

$$\frac{1}{R}[h + \bar{h} + \delta\mathcal{F}_{h,\bar{h}}|_{G_{\mathcal{V}}\to g_{\mathcal{V}}}].\tag{C.15}$$

This is in agreement with the general RG intuition on how the effect of the relevant and irrelevant interaction, specified at a fixed short-distance scale, should scale with the volume.

In our fits in Sec. 4 we accounted for the overall prefactor $1/R$, and then we fit the corrections to the spectrum with $g_{\mathcal{V}}$ as free parameters. We then verified the expected behavior $g_{\mathcal{V}} = G_{\mathcal{V}}/R^{\Delta_{\mathcal{V}}-2}$ with $R$-independent $G_{\mathcal{V}}$.

# D  Ising Hamiltonian on the fuzzy sphere

The fuzzy sphere formulation of the 2D Ising model is based on the study of quantum Hall models on the sphere [75–77]. We consider electrons living on a sphere with a $4\pi s$ monopole at the center. Each electron has spin-1/2 ($\sigma = \Uparrow\Downarrow$) and orbital momentum $m = -s, -s+1, ..., s$ degrees of freedom, where the latter stems from the $2s+1$-fold degeneracy of each lowest landau level (LLL) orbital in this model. Practically, $s$ determines the resolution of the simulation since $N = 2s + 1$ is the total number of orbitals per spin orientation. We consider a half-filled configuration where the number of electrons $N_e$ is equal to $N$.

Following the original construction presented in Ref. [1], the Hamiltonian of the (2+1)D transverse field Ising model on the fuzzy sphere is given by

$$H_{00} = \frac{1}{2}\sum_{m_1,m_2,m=-s}^{s} V_{m_1,m_2,m_2-m,m_1+m}(\mathbf{c}_{m_1}^\dagger\mathbf{c}_{m_1+m})(\mathbf{c}_{m_2}^\dagger\mathbf{c}_{m_2-m}),\tag{D.1}$$

$$H_{xx} = -\frac{1}{2}\sum_{m_1,m_2,m=-s}^{s} V_{m_1,m_2,m_2-m,m_1+m}(\mathbf{c}_{m_1}^\dagger\sigma_x\mathbf{c}_{m_1+m})(\mathbf{c}_{m_2}^\dagger\sigma_x\mathbf{c}_{m_2-m}),\tag{D.2}$$

$$H_{t,z} = -h\sum_{m=-s}^{s}\mathbf{c}_m^\dagger\sigma_z\mathbf{c}_m,\tag{D.3}$$

$$\tilde{H} = H_{00} + H_{xx} + H_{t,z},\tag{D.4}$$

with $\mathbf{c}_m^{(\dagger)} = (c_{m,\uparrow}^{(\dagger)}, c_{m,\downarrow}^{(\dagger)})$ denoting the fermionic annihilation (creation) operators with orbital momentum $m$ and spin orientation $\Uparrow\Downarrow$, $\sigma_{x,z}$ the respective Pauli matrices, $h$ being the transverse field strength and $V_{m_1,m_2,m_2-m,m_1+m}$ parameterizing the interaction on the fuzzy sphere through the Haldane pseudopotentials [75, 78] $V_l$ and the Wigner 3j-symbol $\begin{pmatrix} j_1 & j_2 & j_3 \\ m_1 & m_2 & m_3 \end{pmatrix}$ as

$$V_{m_1,m_2,m_3,m_4} = \sum_{l=0}^{L} V_l(4s-2l+1)\begin{pmatrix} s & s & 2s-l \\ m_1 & m_2 & -m_1-m_2 \end{pmatrix}\begin{pmatrix} s & s & 2s-l \\ m_3 & m_4 & -m_3-m_4 \end{pmatrix}.\tag{D.5}$$

In this work, we restrict ourselves to $L = 1$, such that exclusively $V_0$ and $V_1$ are active while $V_{l>1} = 0$. We remind the reader that the pseudopotentials give a decomposition of any central

potential onto the lowest Landau level. In particular $V_0$ is a projection of the Dirac interaction $\delta(\vec{r} - \vec{r}')$ and $V_1$ the projection of $\nabla^2 \delta(\vec{r} - \vec{r}')$, i.e., they are projections of short-range interactions.

$H_{00}$, up to normal ordering, is the well-studied quantum Hall Hamiltonian. The ground state for the chosen $V_0 \gg V_1$ at $N_e = N$ electrons is a Mott insulator whose spin degrees of freedom form a ferromagnet [79,80]. $H_{xx}$ breaks the SU(2) invariance down to $U(1)$ and the ferromagnet polarizes in the $x$-direction. Finally, $H_{t,z}$ is a polarizing field in the transverse direction and competes with $H_{xx}$. In the limit $h \to \infty$, the spin degrees of freedom form a paramagnet in the $z$-direction. An Ising transition is therefore expected at intermediate $h$ as was shown in Ref. [1]. Note that we performed a rotation of the spin-1/2 degree of freedom around $\sigma^y$ compared to the original Hamiltonian.

Several symmetries leave $H$ invariant. First, it trivially preserves the global $U(1)$-charge symmetry. The SU(2) rotational symmetry on the sphere is also respected, imposing conservation of the momentum and the appearance of degenerate multiplets. $H_{xx}$ and $H_{t,z}$ break the SU(2) spin invariance down to a $\mathbb{Z}_2$ parity conservation of the spin along the $z$ axis:

$$S_z = \sum_{m=-s}^{s} \mathbf{c}_m^\dagger \sigma_z \mathbf{c}_m.$$

We take advantage of the Abelian sector of these three symmetries in our numerical computation. Finally, as we are working at half-filling, there is an additional $\mathbb{Z}_2$ parity-hole symmetry $c_\uparrow^\dagger \leftrightarrow c_\downarrow$ typical from quantum Hall Hamiltonians. Given its non-diagonal nature, we did not implement it explicitly.

# E   Numerical methods

## E.1   Exact diagonalization

Following the discussion in App. D, we take advantage of the U(1) charge-, U(1) momentum- and $\mathbb{Z}_2$ spin-conservation. With this, the symmetry-resolved Hilbert space is constructed as

$$\mathcal{H}_{N_e,\bar{m},p} = \mathrm{Span}\left\{ \prod_{i=1}^{N_e} c_{m_i,\sigma_i}^\dagger |0\rangle \right\}, \tag{E.1}$$

$$\text{with} \quad \sum_{i=1}^{N_e} m_i = \bar{m} \quad \text{and} \quad N_{e,\uparrow} \bmod 2 = p \in \{0,1\}. \tag{E.2}$$

In contrast to studies performed on the torus, the orbital momenta $m_i$ are not folded back to some Brillouin zone upon addition, such that for e.g. $N_e = 2$ and $S_z = 0$ the total orbital momentum can take values $\bar{m} = -2s, -2s+1, ..., 2s$.

In order to find the low-energy spectrum of $H$ from Eq. (D.4) in the Hilbert space of a given symmetry sector, including low-lying excitations, we make use of Arnoldi iteration implemented in the ARPACK [81] package. Similar to Lanczos iteration, the algorithm requires nothing but the implementation of the matrix-vector product $|\phi\rangle = H|\psi\rangle$ to converge a given number of extremal (in this case minimal) eigenvalues. Unfortunately, one of the limiting aspects in terms of reachable Hilbert space dimensions is the integer data type used in the available ARPACK implementation, where the 32-bit nature limited the available system sizes in our simulations. So while $N = 20$ ED simulations are technically feasible with a dimension of about 1.5 billion, because of ARPACK issues the diagonalizations do not seem to converge. Nevertheless, systems sizes of up to $N = 18$ with corresponding maximal (symmetry resolved) Hilbert space dimensions of 113 million could be successfully studied.

## E.2 ITensor MPS implementation

We use ITensors.jl [82,83] to perform our matrix product state computations. Matrix product state computations for fractional Quantum Hall systems have been thoroughly explored [84–88]. The fuzzy sphere is represented by a chain of $2 \times N$ spinless electrons, in ascending order of orbital momentum. The 2 flavor of spins (spin up and down) at a given momentum correspond to consecutive sites. We implemented the U(1) conservation of the electronic charge and the orbital momentum, as well as conservation of the spin parity along the $z$ direction.

The MPO were computed following Eq. (D.4), discarding contributions of amplitude $V_{m_1,m_2,n_2,n_1}$ below $10^{-12}$. They were then compressed using the default compression of ITensors.jl. Note that due to the way ITensors build matrix product operators, we had to decompose the Hamiltonian into several independent Hermitian subHamiltonians including of order 1000 non-zero terms that we recombined before DMRG.

The main algorithm we used to obtain the ground states is two-site DMRG [89]. Due to the large number of symmetries, and in particular the interplay between charge and momentum conservation, two-site updates are not sufficient to ensure the ergodicity of the variational optimization. In fact, if the starting state is a simple local product state in the symmetric basis, it will remain a product state. To remedy to this issue, we initialize our state in an inversion-symmetric product state in the $L^z = 0$ sector. We then apply the Hamiltonian to the product state, and truncate the resulting MPS to a low bond-dimension, before using ladder operators to build an initial state in the targeted symmetry sector. Finally, at each bond dimension $\chi$, we first perform noisy DMRG [90] sweeps to explore phase space followed by conventional noiseless DMRG to converge to the true ground state. The amplitude of the noise was generally kept at $10^{-5}$. Our convergence criterion at a given $\chi$ are variations of the energy and the mid-chain entropy below $10^{-7}$.

To compute excited states, we successively incorporate into the effective Hamiltonian the weighted projectors onto previously-computed low-energy states. This method is quite sensitive to the amplitude of the weights and the precision reached on the low-energy states, but allows us to target a relatively large number of excited states at a modest numerical price. In addition, we took advantage of the SU(2)-invariance of the model to reduce the number of DMRG runs. Namely, we first computed low-energy states at the largest desired $L^z$. Then, we then applied lowering operators to compute the corresponding members of the multiplets at smaller $L^z$ instead of starting DMRG from scratch. In practice, we found that we could reliably converge around 10 eigenstates in the largest symmetry sector, before the combination of lack of precision and ergodicity prevented us to reach our desired accuracy. We also note that entropy increases quickly with the energy levels, which contributes to the reduced precision. We investigated whether including the $L^2$ operator into the Hamiltonian in order to penalize states with $L^z < L$ could simplify the computation. We nonetheless found that our first approach was generally more precise.

As a final comment, the fact that the system lives on a sphere implies that the mid-chain entropy for the ground state, even for a gapped model, grows as $\sqrt{N}$ instead of $O(1)$. Indeed, the area of the interface between the two orbitals close to the hemisphere is proportional to the radius of the sphere. At the same time, the number of orbitals is formally proportional to the magnetic flux going through the surface of the sphere. Fixing the magnetic length as our unit, this means that the surface of the sphere is, up to irrelevant prefactors, exactly $N$, and therefore its radius is proportional to $\sqrt{N}$. The scaling of the entropy as $\sqrt{N}$ is the main drawback and limitation of tensor network computations on a (fuzzy) sphere.

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
