# Peer review of "Exact Diagonalization, Matrix Product States and Conformal Perturbation Theory Study of a 3D Ising Fuzzy Sphere Model"

_SciPost Physics, doi:SciPost Phys. 19, 076 (2025)_

## Round 1 · Referee Report · Anonymous (Referee 2) · 2025-7-15

Report

In this paper the authors apply the methods of conformal perturbation theory (CPT) to study the data numerically generated by the fuzzy sphere realization of 2+1d Ising universality. They also have a nice warm-up showcasing the method in the case of 1+1d lattice spin model, also near Ising universality.

In the lower dimensional case, where the system is under more control due to the integrability of the Ising CFT, the authors are able to go to higher order in perturbation theory and achieve impressive agreement between CPT and numerical data. They also include the effect of the leading irrelevant operators, which further improves the results.

In the case of 2+1d, the authors mostly rely on linear order in CPT, as less is known about the 3d Ising CFT data. Here also, the numerics are more expensive and so the effective system size is smaller, resulting in more irrelevant operators potentially playing a role. Despite the additional complications, the authors find that they are able to fit the numerics well using CPT, in particular for some low-energy states. They also have a nice analysis elucidating the original choice of Haldane pseudopotential coefficient $V_0 =4.75$ by the original fuzzy sphere authors. Indeed, they show that for this choice, the leading irrelevant Wilson coefficients happen to be small, explaining how they were able to agree with the bootstrap despite not having a very large sphere.

Finally, they use CPT to extract some OPE coefficients. Some of these show decent agreement with previous known results, and others are new determinations.

Interestingly, the authors were not able to further systematically improve the agreement with CPT by including the effect of leading irrelevant operators. This is perhaps because there are many irrelevant operators which might play a role given current numerical reach, and nailing all their Wilson coefficients is more involved.

I believe this paper is well written, contains solid analysis and is worthy of publication. I only have minor questions regarding some of the choices the authors make, when they choose to focus on some irrelevant operators, while excluding others:

Firstly, in Fig. 11 they choose to look at $g_C$ and not at $g_{T’}$ (which has a similar dimension). Is this just because they were looking at $\sigma$ and descendant states (as opposed to $\epsilon$ and its descendants)? Do they expect that $g_C$ is indicative of a trend, so that $g_{T’}$ will also be small for $V_0=4.75$?

Secondly, in section 7 about mixing, why do they consider the effect of $g_{\epsilon’’}$ before considering the effects of $g_C$ and $g_{T’}$, both of which are more relevant?

Recommendation

Publish (easily meets expectations and criteria for this Journal; among top 50%)

  • validity: -
  • significance: -
  • originality: -
  • clarity: -
  • formatting: -
  • grammar: -

Author:  Loïc Herviou  on 2025-08-20  [id 5744]

(in reply to Report 2 on 2025-07-15)
Category:
answer to question

Dear Referee,

Thank you for your positive assessment of our paper. You can find below a detailed answer for your questions.

Best regards, The authors.

1) « Firstly, in Fig. 11 they choose to look at $g_C$ and not at $g_{T'}$ (which has a similar dimension). Is this just because they were looking at $\sigma$ and descendant states (as opposed to $\varepsilon$ and its descendants)? Do they expect that $g_C$ is indicative of a trend, so that $g_{T'}$ will also be small for $V_0=4.75$? »

In our explorations, we have not achieved reliable fits including $T'$ and we do not report those fits. We added footnote 20 to stress that we do not include $T'$ and that this should be done in the future.

2) « Secondly, in section 7 about mixing, why do they consider the effect of $g_{\varepsilon''}$ before considering the effects of $g_C$ and $g_{T'}$, both of which are more relevant? »

That section is exploratory in spirit. What the referee suggests is definitely worth trying in future work. We modified the last paragraph of Section 7 to mention this.

---

## Round 1 · Referee Report · Anonymous (Referee 1) · 2025-7-15

Report

This paper studies the Fuzzy Sphere regularization through the lense of conformal perturbation theory (CPT). The fuzzy sphere technique consists in realizing a given universality class via electrons moving under the influence of a magnetic field and of a potential. The symmetries of the potential prescribe the universality class. The magnetic field plays the role of a regulator: the Hilbert space of the theory is truncated to the lowest Landau level, thus removing high energy states which are explored by the system, unless the magnetic field, and the gap to the next Landau level, diverge. As with any truncation, finite size effects drive the system off criticality, and a crucial part of the task is to minimize such effects by tuning the coefficients in the potential.

The authors use CPT to parametrize deviations from criticality via relevant and irrelevant CFT operators. Then, they match these couplings to the coefficients of the Fuzzy Sphere Hamiltonian by fitting data from exact diagonalization to the CPT predictions. This step requires accurate knowledge of the low lying CFT states and of their OPE coefficients, which for the 3d Ising model is provided by the conformal bootstrap. CPT then becomes fully predictive and allows to extract finite size corrections for (in principle) any other measured quantity.

The procedure has a good degree of success in explaining the Fuzzy sphere data, although discrepancies remain and are duly noted in the text and figures. Overall, the paper is pedagogical,
detailed, and useful, and certainly meets the criteria for publication on SciPost. The only real weakness of the results is that they do not seem to provide a decisive improvement in tuning the model, rather confirming the accuracy of the tuning already performed in the seminal work on the subject. Nevertheless, CPT does provide conceptual clarity, and the promise of more systematic improvements in the future.

I will be happy to recommend the paper for publication, after the authors consider some minor observations reported below.

Requested changes

1- At the end of section 2.2, it is stated that the number of electrons N can be identified as the volume of the sphere, i.e. the finite size parameter. While the half filling condition does link the large N limit to the thermodynamic limit, the specific relation to the volume is not motivated explicitly, and yet plays a crucial role in what follows. It would be worth expanding the explanation about this point, and possibly adding rigour to it. 2 - In going from eq. 16 to 17, it is assumed that the perturbing operator is invariant under the Laplace transform. This seems to require some restriction on the nature of the CFT operator, since there are cases (e.g. F^2 in Maxwell theory) where the presence of time derivatives of the elementary field produces a change. In the pedagogical spirit of this section, it would be nice if the authors could add a sentence explaining what restriction is made here. 3 - In eq. 32, it would be useful to specify that J is positive. 4 - In line 426, it is spoken about a mode with momentum 2\pi/R. Is this a typo? The confusion stems from the fact that momenta are usually quantized to be integer multiples of 1/R . 5 - In the same section the authors explain that the global Z2 symmetry is realized as translation by one lattice spacing. Then, Z2 odd states must have momenta which are odd multiples of \pi/a (a being the lattice spacing). These momenta only exist for even number of spins with periodic boundary conditions. Are Z2 odd state only accessible in this case? If useful, the authors might add an explanation (or simply satisfy the referee's curiosity in their reply). 6 - In section 5.4., the authors report on the difficulties in explaining some deviations of the energy levels from the CFT prediction by using CPT. While this is fine, some of the final considerations in the paragraph seem a bit vague ("reduce deviation when it is significant is easier than to improve agreement further when it is already small"), at times tautological ("some small effects are at play [...] like level mixing [...] or perhaps something else"). 7 - Fig. 11 shows that the dependence of CPT couplings on N dictated by dimensional analysis is not always observed. While this fact highlights the importance of explaining the relation between N and R as requested in point 1, it also suggests the question if the dependence of the effective Hamiltonian on the cut-off might be the subject of a more detailed analysis via the renormalization group. For other truncation schemes (e.g. Hamiltonian truncation) a rather detailed understanding was reached as to how the couplings run with the cut-off: it is unclear to the referee if such understanding is available in the Fuzzy sphere case, and if it is expected to modify the analysis discussed in the paper. One might wonder what happens if one explicitly integrates out a monopole harmonic, and the referee would be grateful for a clarification on this point.

Recommendation

Ask for minor revision

  • validity: high
  • significance: high
  • originality: good
  • clarity: top
  • formatting: excellent
  • grammar: perfect

Author:  Loïc Herviou  on 2025-08-20  [id 5743]

(in reply to Report 1 on 2025-07-15)
Category:
remark
answer to question
correction

Dear Referee,

Thank you for your positive report. Here follows a detailed answer for your comments.

The authors.

1) « At the end of section 2.2, it is stated that the number of electrons N can be identified as the volume of the sphere, i.e. the finite size parameter. While the half filling condition does link the large N limit to the thermodynamic limit, the specific relation to the volume is not motivated explicitly, and yet plays a crucial role in what follows. It would be worth expanding the explanation about this point, and possibly adding rigour to it. » 

We assumed the referee meant to indicate the end of section 2.3. We clarified the discussion, which we summarize below.

The key point is that the area of the sphere scales as the number of magnetic orbitals (spin degeneracy not counted). This choice of scaling correspond to a physically meaningful limit of constant magnetic field through the sphere when scaling $N$. It then just happens to match the number of electrons in our model as the Ising criticality is obtained at half-filling.

2) « In going from eq. 16 to 17, it is assumed that the perturbing operator is invariant under the Laplace transform. This seems to require some restriction on the nature of the CFT operator, since there are cases (e.g. $F^2$ in Maxwell theory) where the presence of time derivatives of the elementary field produces a change. In the pedagogical spirit of this section, it would be nice if the authors could add a sentence explaining what restriction is made here. »

No such restriction is made here. We put a sentence and footnote after Eq. (17) to explain this.

3) « In eq. 32, it would be useful to specify that J is positive. »

We added that $J>0$ below Eq. 32.

4) « In line 426, it is spoken about a mode with momentum $2\pi/R$. Is this a typo? The confusion stems from the fact that momenta are usually quantized to be integer multiples of 1/R . »

We thank the referee for noticing this typo. The momentum is indeed $\frac{2\pi}{N} = \frac{1}{R}$.

5) « In the same section the authors explain that the global Z2 symmetry is realized as translation by one lattice spacing. Then, Z2 odd states must have momenta which are odd multiples of $\pi/a$ (a being the lattice spacing). These momenta only exist for even number of spins with periodic boundary conditions. Are Z2 odd state only accessible in this case? If useful, the authors might add an explanation (or simply satisfy the referee's curiosity in their reply). »

The relationship between $\mathbb{Z}_2$-odd states and translation is more complex than our sentence implied. $\mathbb{Z}_2$-odd states exist in the Hilbert space even if the length of the chain is odd. Due to incompatible boundary conditions, they may have additional corrections, and typically will match what is expected from a different set of boundary conditions for continuous states (think about the difference between Ramond and Neveu–Schwarz sectors in a conformal field theory). Typically, one can recover correct scaling for odd-length chains by playing with antiperiodic boundary conditions instead. This is a well-known complexity when trying to match spectra, and it is more convenient to discard it by working directly with even (or $0 \mod 4$) chains. We modified the text to specify even-length chains, but did not develop more in the main text.

6) « In section 5.4., the authors report on the difficulties in explaining some deviations of the energy levels from the CFT prediction by using CPT. While this is fine, some of the final considerations in the paragraph seem a bit vague ("reduce deviation when it is significant is easier than to improve agreement further when it is already small"), at times tautological ("some small effects are at play [...] like level mixing [...] or perhaps something else"). »

We agree that the phrasing was imperfect. We have streamlined the last sentences of Sec. 5.4 to address this.

7) « Fig. 11 shows that the dependence of CPT couplings on N dictated by dimensional analysis is not always observed. While this fact highlights the importance of explaining the relation between N and R as requested in point 1, it also suggests the question if the dependence of the effective Hamiltonian on the cut-off might be the subject of a more detailed analysis via the renormalization group. For other truncation schemes (e.g. Hamiltonian truncation) a rather detailed understanding was reached as to how the couplings run with the cut-off: it is unclear to the referee if such understanding is available in the Fuzzy sphere case, and if it is expected to modify the analysis discussed in the paper. One might wonder what happens if one explicitly integrates out a monopole harmonic, and the referee would be grateful for a clarification on this point. »

In Hamiltonian truncation (HT), one starts with a UV complete model and tries to construct an effective Hamiltonian by integrating out modes above the UV cutoff. Invariably, effective Hamiltonian computation in HT is carried out by taking advantage of weak coupling at the cutoff scale.

In our problem, the model is strongly coupled and it RG-flows to a strongly coupled CFT. It looks challenging to us to set up a controlled RG calculation which would compute the effective Hamiltonian near the end point of the RG flow. Were we to integrate a monopole harmonic, this would result in a perturbation of the microscopic Hamiltonian, which would still have to be translated to a perturbation of the CFT Hamiltonian. Unfortunately, we see no controlled way of doing this.

We thank the referee for the inquiry, but detailed exploration of this line of thought looks like it is beyond the scope of our paper, so we have not implemented any changes at present.

---

## Round 1 · Referee Report · Anonymous (Referee 3) · 2025-7-16

Report

This is an interesting paper whose main results are - a precise analysis of the Ising Model on the Fuzzy Sphere through the lens EFT methods around the IR CFT, - a new proposal to measure OPE coefficients based on EFT fits. Furthermore, the review of section 2-4 + appendices, will provide a useful resource to many.
The paper is generally well written.

I do not have any major criticism. However here are few comments or suggestions that may improve the paper:

  • The discussion of Fig5 in sec 4: I found the caption slightly confusing for the following reasons:
dashed lines are hard or barely visible: are they always behind solid lines? why? I would have expected the lines corresponding to the fit to agree with CFT for a broader range. Similarly, are circles always inside squares?
what is the Rimicke model being used here, the full non-linear CPT fit used afterward in the discussion?
There are no explanations about these features in the main text. The only reference being made to the figure is to argue that the spectrum broadly agrees with CFT at hx=0.803.

Sec 5.3.

  • I do not fully understand the logic flow at the beginning of the paragraph “This complex structure…”
The sentence is not a sharp assertion, since it only states ‘.. delicate to justify …’. Nevertheless, what the sentence suggests goes against the main point of the work: deviations of fuzzy ED are intricate in (roughly) the same precise way as CPT is.
The paragraph also raises two other points of challenge: scarce CFT data for CPT, and presence of quasi degeneracies.
I guess that the authors are trying to say that despite these challenges they will manage to perform the CPT analyses.
  • It is said that, descendants at small/large spin are more strongly affected by perturbations with small/large spin. However the plots presented in fig. 9 show a maximum when varying along the y-direction, thus not exactly the linear correlation that seems to be implied by the previous sentence.
  • A couple of comments in the last paragraph: 1.- in view of the first sentence two paragraphs before “This complex structure…”, the first sentence of the last paragraph is somewhat misleading. It may seem to imply that only g_Eps' and g_C are small while other couplings are big. I believe the authors mean that typically the couplings g_C and g_Eps' are the largest among the irrelevant contributions, but for the specific choice V0=4.75 these two become small relative to further irrelevant contributions. 2.- It seems that the sentence “Moving V0…” should come before “Changing h…”.

Sec 5.4

  • “The left most panel … its first THREE descendants ….” —> “The left most panel … its first FOUR descendants ….”
Or “The left most panel displays $sigma$, $\partial \sigma$, and the next three descendants.”
Then the next sentence also needs adjustments: “The second panel does the same for epsilon…”.
  • It is said that $\partial^2 \epsilon$ is shifted vertically, which is true. But there are many other states showing larger shifts (i.e. broad shaded regions), thus it is unclear why this is singled out.
  • why the authors do not show the value of g_epsilon as a function of h for the minimal CPT?
Is this because the result is very similar to the first plot of the second raw in fig. 11? If not, could they explain how does it look like?

Sec 5.5:

  • it is interesting. Would have been interesting to see large N extrapolations, and determinations of CPT data.
  • I enjoyed the speculations to justify the drift or dependence h_zero(N) determined via g_epsilon(h_zero)=0.

Section 6:

  • perhaps I misunderstood the idea proposed around eq. (39).
The method has access to ratios of OPE coefficients, since one can always renormalise gEpsilon by say f{epsilon\sigma\sigma}.
Do the authors agree? Is this the reason why the value of f{epsilon\sigma\sigma} looks exactly flat in fig. 12?
If true, I did not find this clearly stated that those points do not constitute an actual test.
  • btw, CB , i.e. conformal bootstrap, has never been defined.

To conclude, after addressing my questions, I will be able to recommend the publication of this very interesting work.

Recommendation

Publish (surpasses expectations and criteria for this Journal; among top 10%)

  • validity: -
  • significance: -
  • originality: -
  • clarity: -
  • formatting: -
  • grammar: -

Author:  Loïc Herviou  on 2025-08-20  [id 5746]

(in reply to Report 3 on 2025-07-16)
Category:
remark
answer to question
correction

Dear Referee,

Thank you for your positive report. Please find below the answer to your questions.

Best regards, The authors

1) « The discussion of Fig5 in sec 4: I found the caption slightly confusing for the following reasons: dashed lines are hard or barely visible: are they always behind solid lines? why? I would have expected the lines corresponding to the fit to agree with CFT for a broader range. Similarly, are circles always inside squares? what is the Rimicke model being used here, the full non-linear CPT fit used afterward in the discussion? There are no explanations about these features in the main text. The only reference being made to the figure is to argue that the spectrum broadly agrees with CFT at hx=0.803. »

Indeed, the dashed lines are almost always behind the solid lines, the circles are inside the squares, and the full Reinicke model is used here. We have expanded the caption to make this clear, and added a sentence at the end of Section 4.

2) « Sec 5.3.: I do not fully understand the logic flow at the beginning of the paragraph "This complex structure...".The sentence is not a sharp assertion, since it only states ".. delicate to justify...". Nevertheless, what the sentence suggests goes against the main point of the work: deviations of fuzzy ED are intricate in (roughly) the same precise way as CPT is. The paragraph also raises two other points of challenge: scarce CFT data for CPT, and presence of quasi degeneracies. I guess that the authors are trying to say that despite these challenges they will manage to perform the CPT analyses. »

We rewrote the last three paragraphs of Section 5.3 to make our logic more clear.

3) « Sec 5.3: It is said that, descendants at small/large spin are more strongly affected by perturbations with small/large spin. However the plots presented in fig. 9 show a maximum when varying along the y-direction, thus not exactly the linear correlation that seems to be implied by the previous sentence. »

We did not mean to imply a simple linear correlation, merely that, as can be seen by the shading, higher-spin descendants tend to be more affected by higher-spin perturbations than the lower-spin descendants (see Fig. 9, top right for the clearest example). In particular, indeed, the maximally coupled does not seem to shift linearly with the spin of the descendant, though we did not perform a sufficiently thorough analysis to conclude rigorously (the higher the spin, the larger the finite-size effects). We clarified our statement in the main text.

4) « Sec 5.3: A couple of comments in the last paragraph: 1.- in view of the first sentence two paragraphs before “This complex structure…”, the first sentence of the last paragraph is somewhat misleading. It may seem to imply that only $g_\varepsilon'$ and $g_C$ are small while other couplings are big. I believe the authors mean that typically the couplings $g_C$ and $g_\varepsilon'$ are the largest among the irrelevant contributions, but for the specific choice $V_0=4.75$ these two become small relative to further irrelevant contributions. 2.- It seems that the sentence “Moving V0…” should come before “Changing h…”. »

We rewrote the last three paragraphs of Sec. 5.3, to make the logic more clear.

5) « Sec 5.4: "The left most panel … its first THREE descendants ..." —> “The left most panel … its first FOUR descendants ….”. Or "The left most panel displays sigma, $\partial \sigma$, and the next three descendants." Then the next sentence also needs adjustments: “The second panel does the same for epsilon…”. »

We thank the referee for spotting this typo which we corrected.

6) « Sec:5.4: It is said that $\partial^2\varepsilon$ is shifted vertically, which is true. But there are many other states showing larger shifts (i.e. broad shaded regions), thus it is unclear why this is singled out. »

There is indeed nothing special about $\partial^2\varepsilon$. We corrected the corresponding phrase to remove reference to specific energy levels.

7) « Sec 5.4: why the authors do not show the value of $g_\varepsilon$ as a function of h for the minimal CPT? Is this because the result is very similar to the first plot of the second raw in fig. 11? If not, could they explain how does it look like? »

Yes, the not shown plot is very similar. We added footnote 19 on p. 23 to state that.

8) « Sec 5.5: it is interesting. Would have been interesting to see large N extrapolations, and determinations of CPT data. »

Thank you for a supportive comment. Some of the hints we see in this section are tantalizing, but some behaviors as we described need to be understood better before reliable extrapolation can be performed. We hope to come back to this in future work. For the moment we leave the section as is.

9) « Sec: 5.5 I enjoyed the speculations to justify the drift or dependence $h_0(N)$ determined via $g_\varepsilon(h_0)=0.$ »

Thank you for a supportive comment.

10) « Section 6: perhaps I misunderstood the idea proposed around eq. (39).The method has access to ratios of OPE coefficients, since one can always renormalise gEpsilon by say $f_{\varepsilon\sigma\sigma}$. Do the authors agree? Is this the reason why the value of $f{epsilon\sigma\sigma}$ looks exactly flat in fig. 12? If true, I did not find this clearly stated that those points do not constitute an actual test. btw, CB , i.e. conformal bootstrap, has never been defined. »

Yes and no. "No" because our normalization of $g_\epsilon$ is in principle fixed via eqs. (17), (19), using furthermore the fact that the CFT operators such as $\epsilon$ have normalization conventionally fixed via a unit-normalized two-point correlation function. "Yes" because to extract $g_\epsilon$, which in this analysis is done via a fit to $\sigma$ and $\partial\sigma$ level (the fit which also determines the speed of light), we assume $f_{\sigma\sigma\epsilon}$ fixed to its CB value. This explains why the two corresponding lines in Fig.12 are completely flat. This was in principle stated in the phrase "Based on our prescription, the OPEs for $\sigma$ and $\partial\sigma$ are the ones we used to determine $g_\varepsilon$ in the first place, and hence identically agree with CFT predictions. All the other OPE estimates reported in this figure however are independent ED data." However, to stress this further, we added a paragraph below Eq.~(41).

We also added the definition of CB.

Anonymous on 2025-09-05  [id 5785]

(in reply to Loïc Herviou on 2025-08-20 [id 5746])

The authors addressed all my questions.
I recommend the work to be published.

---

## Round 2 · Author Response

Dear editors of SciPost,

We thank the referees for their very positive assessment of our work.
We took into account their remarks — the different changes are listed below. We also provide a direct answer to each referee as a comment to their report.

We updated the bibliography, including journal information to cited works on the arXiv which got published in the meantime.
Changes in the manuscripts are in blue.

Best regards,
The authors.

---

## Round 2 · List of Changes

• A sentence and a footnote after Eq. (17).
  • Section 2.3. Clarified the relationship between the electron number $N$ and $R$.
  • Added $(J>0)$ below Eq. 32.
  • Sec 4: corrected the definition of the momentum, and specified that we consider even-length chains.
  • Sec 4: clarified the discussion on the effect of the spin in CPT.
  • Sec. 4: expanded the caption of Fig.5 and added a sentence at the end of the section.
  • Sec. 5.3 the last three paragraphs rewritten.
  • Sec 5.4: corrected a typo in the description of Fig. 10.
  • Sec. 5.4: added footnote 19.
  • Sec. 5.4: the last two sentences rewritten.
  • Sec. 6: paragraph below Eq. (41).
  • Sec 7 last paragraph modified.

---

## Editorial Decision

published